# UNIVERSAL SLEEP DECODER: ALIGNING AWAKE AND SLEEP NEURAL REPRESENTATION ACROSS SUBJECTS

## ABSTRACT

Decoding memory content from brain activity during sleep has long been a goal in neuroscience. While spontaneous reactivation of memories during sleep in rodents is known to support memory consolidation and offline learning, capturing memory replay in humans is challenging due to the absence of well-annotated sleep datasets and the substantial differences in neural patterns between wakefulness and sleep. To address these challenges, we designed a novel cognitive neuroscience experiment and collected a comprehensive, well-annotated electroencephalography (EEG) dataset from 52 subjects during both wakefulness and sleep. Leveraging this benchmark dataset, we developed the Universal Sleep Decoder (USD) to align neural representations between wakefulness and sleep across subjects. Our model achieves up to 16.6% top-1 zero-shot accuracy on unseen subjects, comparable to decoding performances using individual sleep data. Furthermore, fine-tuning USD on test subjects enhances decoding accuracy to 25.9% top-1 accuracy, a substantial improvement over the baseline chance of 6.7%. Model comparison and ablation analyses reveal that our design choices, including the use of (i) an additional contrastive objective to integrate awake and sleep neural signals and (ii) the pretrain-finetune paradigm to incorporate different subjects, significantly contribute to these performances. Collectively, our findings and methodologies represent a significant advancement in the field of sleep decoding.

## 1 INTRODUCTION

Sleep plays a fundamental role in memory consolidation (Klinzing et al., 2019; Brodt et al., 2023). Past memories are known to reactivate during sleep, especially during the N2/3 stage of non-rapid eye-movement (NREM) sleep (Ngo & Staresina, 2022). In rodents, hippocampal cells have been found to replay their firing patterns during sleep, recapitulating awake experiences in a time-compressed order (Wilson & McNaughton, 1994; Skaggs & McNaughton, 1996). In humans, while direct cell recordings are rare, recording scalp electroencephalograms (EEG) during sleep is possible. Recent work on human sleep decoding has identified endogenous memory reactivation during the N2/3 stage of sleep, the extent of which was positively related to subsequent memory performance (Schreiner et al., 2021).

Despite the significance of sleep decoding in humans, attempts of this sort is scarce in both neuroscience and computer science community. This is because there does not exist a well-annotated sleep dataset that provides clear ground-truth information about which memory is activated and when during sleep. The population neural activity during wakefulness and sleep differ greatly, causing classifiers trained on awake periods to struggle when applied to sleep states (Liu et al., 2022). Generalizing neural representation across subjects is especially challenging during sleep due to the spontaneous nature of memory reactivation without timed neural responses.

To capture the content of neural reactivation in humans, we designed an innovative cognitive neuroscience experiment based on the classical targeted memory reactivation (TMR) paradigm (Rasch et al., 2007), see Fig.1 for more details. We employed a closed-loop stimulation system allowing for real-time, automatic sleep staging (Vallat & Walker, 2021). When subjects reached the N2/3 stage of NREM sleep, auditory cues paired with visual objects were played every 4-6 seconds, with concurrent whole-brain EEG recordings. This approach provided precise timing and content of memory reactivation during sleep, facilitating the training of a neural decoder on these cued sleep intervals.

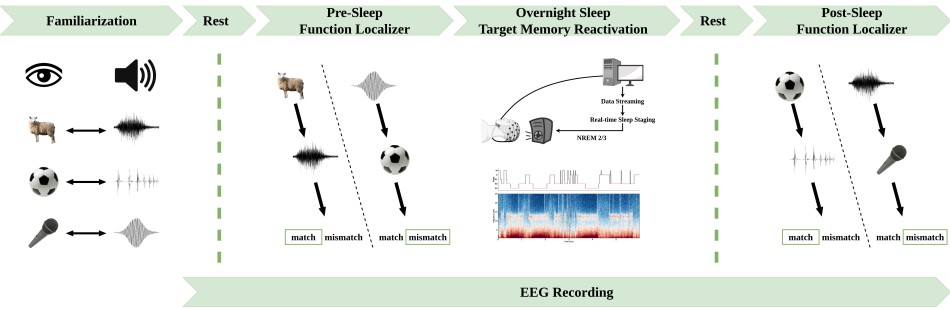

Figure 1: **The experimental design for sleep decoding.** Before the experiment began, subjects were instructed to memorize 15 predefined pairs of pictures and sounds, each sharing the same semantic meaning; for example, a picture labeled "sheep" was paired with the sound of a sheep crying. After a rest period, subjects were exposed to 1000 trials, each of which contains one picture and one sound selected from these 15 pictures and 15 sounds (the picture and sound in one trial is not required to be paired), presented in either image-audio or audio-image order, and were asked to determine whether the pairs corresponded correctly. Subsequently, during overnight EEG recording, online, real-time sleep staging was performed. When subjects entered an NREM 2/3 sleep stage, auditory cues — randomly selected from the image-audio pairs — were played every 4-6 seconds. This step was crucial as it provided ground truth regarding which memory was activated and when. Following overnight sleep, subjects were again presented with 1000 image-audio pairs. Whole-brain 64-channel EEG recordings were collected throughout the experiment, during both sleep and wakefulness. More details can be found in Appendix.A.

Before and after sleep, subjects were asked to recall the auditory cue-visual object pairs, enabling the alignment of neural representations elicited by the same cue during both sleep and wakefulness. This effort yielded a comprehensive dataset from 52 subjects, serving as the benchmark dataset for developing sleep decoders.

Based on this dataset, we introduce a Universal Sleep Decoder (USD) capable of decoding neural signals during NREM sleep, even on unseen subjects in a zero-shot manner. USD was pretrained in a supervised manner across a pool of subjects, learning subject-agnostic features and offering off-the-shelf decoding capabilities for new subjects. Given the challenges of sleep data collection, we also explored the potential to enhance sleep decoding performance by integrating relatively abundant awake task data. To encourage USD to learn domain-agnostic features, we incorporated a contrastive loss to align the neural representations elicited by the same cue during both sleep and wakefulness. With these design choices, USD achieves up to $16.6\%$ top-1 zero-shot accuracy on unseen subjects – comparable to single-subject sleep decoding performances using individual sleep data. Furthermore, fine-tuning USD on test subjects boosts decoding accuracy to $25.9\%$ top-1 accuracy, a notable improvement over the baseline chance of $6.7\%$.

It is noteworthy that the proposed USD can also be effectively applied to other types of brain recordings with high temporal resolution, e.g., magnetoencephalography (MEG) or stereo-electroencephalography (sEEG). Moreover, as our method does not require the averaged signal across trials (Schönauer et al., 2017), it can be extended for real-time sleep decoding, providing a powerful tool for manipulating memory reactivation in real time during sleep.

Our contributions are:

1. Establishing a well-annotated sleep dataset in humans, with ground truth regarding which memory was activated and when.

2. Showing that neural activity during wakefulness shares representations with sleep and aligning these can enhance the efficacy of sleep decoding.

3. The Universal Sleep Decoder (USD), a reusable, off-the-shelf, subject-agnostic model, offering a degree of zero-shot capability across subjects and can further improve performance through fine-tuning, establishing a new standard in sleep decoding methodologies.

## 2 RELATED WORK

### 2.1 SLEEP DECODING

During sleep, particularly during NREM sleep, humans are largely unconscious, and neural reactivation occurs spontaneously (Siclari et al., 2017; Schönauer et al., 2017). Consequently, gathering a well-annotated dataset that offers precise timing and content of neural reactivation during sleep is challenging, posing a significant hurdle in sleep decoding research. To address this challenge, some studies have attempted to extract memory reactivation content by soliciting reports from subjects either after they awaken or during their lucid dreaming (Horikawa et al., 2013; Siclari et al., 2017; Konkoly et al., 2021; Dresler et al., 2012). However, the data obtained is far less than what is required to train the model, resulting in the current sleep decoder being trained on data from wakeful periods (Horikawa et al., 2013). Furthermore, this approach is limited solely to the REM sleep stage, where neural representation resembles that of wakefulness. Considering the neural patterns during NREM sleep—a period associated with memory replay—these exhibit even greater differences from those observed during wakefulness, making the memory decoding substantially more intricate. Consequently, this approach is unsuitable for decoding during NREM sleep.

Other studies in this field directly ignore the fact that the subjects are asleep (Türker et al., 2022). Sleep has classically been considered as a time when we are disconnected from the world, with significantly reduced (or absent) reactivity to external stimuli. However, several studies in recent years indicate that sleepers can process external stimuli at different levels of cognitive representation, encompassing semantic and decision-making stages, rather than merely at the level of low-level sensory processing (Strauss et al., 2015; Issa & Wang, 2011; Kouider et al., 2014). Furthermore, learning-related sensory cues presented during sleep positively impact subsequent recall of cue-related material upon awakening (Rasch et al., 2007; Hu et al., 2020), which is commonly referred to as Target Memory Reactivation (TMR). We follow these studies to design our cognitive neuroscience experiment, see Appendix.A for more details.

### 2.2 CONTRASTIVE LEARNING

Recently, the field of unsupervised learning has grown rapidly with the powerful contrastive learning (Chen et al., 2020). Meanwhile, Khosla et al. (2020) extend the contrastive loss to the supervised setting, thus allowing the model to learn a more discriminative representation for the classes. Along with the recent development of large models (Baevski et al., 2020; Radford et al., 2021), many studies exploit the similarity of neural representations between the human brain and large models to improve the performance of neural decoding (Défossez et al., 2022; Chen et al., 2023). However, these approaches often require the collection of a dataset with a considerable number of labels for a single subject, which is impossible for the experiment design of sleep decoding (Türker et al., 2022; Hu et al., 2020). Similar to the TMR experiment setup, our designed experiment only allows us to gather a dataset with few labels, primarily because of the limited number of image-audio pairs available during the familiarization stage (see Fig.1).

Given the scarcity of available sleep datasets, no previous study has attempted to use contrastive learning to bridge the gap between brain activities recorded during wakefulness and sleep (i.e., two domains, in terms of domain generalization (Wang et al., 2022a)). The most relevant studies to our work mainly come from the fields of computer vision (Kim et al., 2021; Wang et al., 2022b) and natural language processing (Peng et al., 2018). Similar to these studies, we employ contrastive learning to align the neural representations elicited by the same cue during both wakefulness and sleep, thereby facilitating the acquisition of domain-agnostic features. Besides, domain adversarial training (Ganin et al., 2016; Zhu et al., 2020) could potentially offer a more robust approach for acquiring domain-agnostic features. However, that approach is not extensively investigated within the scope of this work.

### 2.3 PRETRAIN-FINETUNE PARADIGM

The pretrain-finetune paradigm is widely used in computer vision (Krizhevsky et al., 2012; He et al., 2022) and natural language processing (Devlin et al., 2018). In the early stages of pretraining development, supervised pretraining approaches (Krizhevsky et al., 2012) often outperform unsupervised

pretraining approaches (Zhang et al., 2016; Noroozi & Favaro, 2016; Pathak et al., 2016) and serve as a baseline to evaluate the effectiveness of unsupervised methods.

Limited by the amount of available EEG recordings, only a few studies explored unsupervised pretraining methods for EEG signals (Kostas et al., 2021; Bai et al., 2023; Li et al., 2022b). Recently, most of the studies seek supervised pretraining methods to integrate different subjects (Zhao et al., 2021; Sun et al., 2022), learning subject-agnostic features that are generalizable to new subjects. Since there are fewer publicly available EEG recordings during sleep compared to those during wakefulness, applying unsupervised pretraining methods for sleep decoding is still under exploration. Consequently, we adopt supervised pretraining methods to further improve the performance of sleep decoding.

# 3 METHOD

We first formalize the general task of sleep decoding. Then, we introduce the deep learning architecture, and motivate the use of contrastive loss for training. Finally, we introduce the pretrain-finetune paradigm when training on multiple subjects.

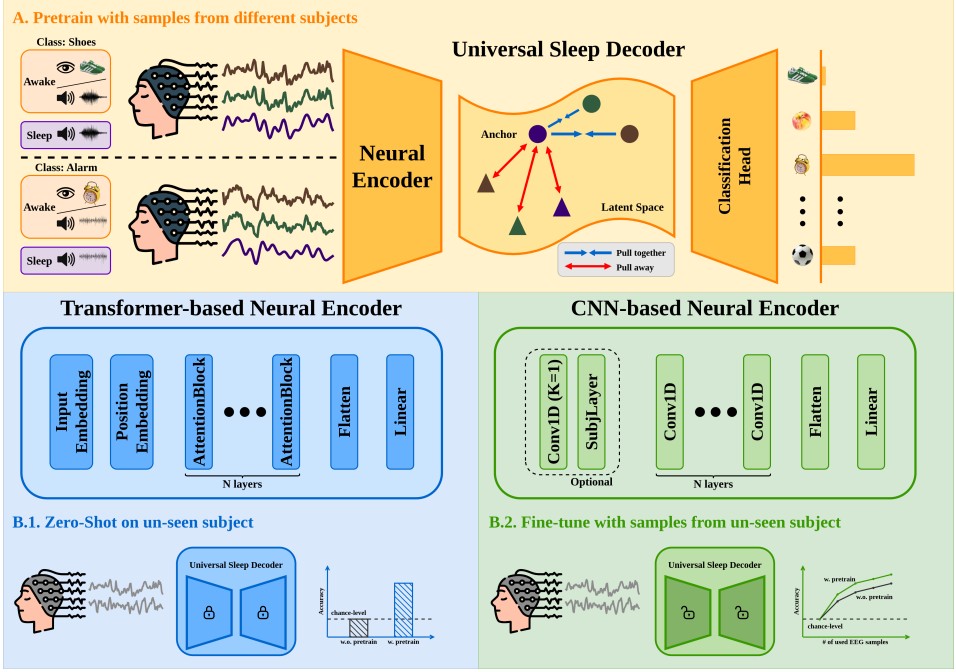

Figure 2: **Overview of Universal Sleep Decoder (USD).** Our method comprises two main components: 1) supervised pretraining across multiple subjects, 2) model evaluation on unseen subjects either with or without finetune. **(A). Pretrain stage.** The model architecture consists of the neural encoder and the classification head. Contrastive loss is used to regularize the latent space, encouraging similarity among features sharing the same semantic class but coming from different domains (e.g. $\{(\mathcal{D}_i^{img}, \mathcal{D}_i^{aud}, \mathcal{D}_i^{tmr})\}$) of the same subject $i$. **(B.1). Zero-Shot Evaluation.** Transformer-based neural encoder is mainly designed for zero-shot evaluation. **(B.2). Fine-tune Evaluation.** CNN-based neural encoder is mainly designed for fine-tune evaluation.

## 3.1 PRELIMINARIES

Following our designed cognitive experiment paradigm (see Fig.1), we recorded three different kinds (i.e., image-evoked, audio-evoked, and TMR-related) of EEG signals from each subject. Note that we refer to audio-evoked EEG signals during sleep as TMR-related, instead of TMR-evoked, as our experiment differs from the TMR paradigm. Different kinds of EEG signals can be viewed as different domains due to the large gap in their neural patterns, especially between awake and

sleep EEG signals (Schönauer et al., 2017). As they share the same semantic classes, we define a pair-wise dataset format $\mathcal{D} = \{(x_n, y_n)\}_{n=1}^N$ and its corresponding single EEG data format $\mathcal{X} = \{x_n\}_{n=1}^N$, where $x \in \mathbb{R}^{C \times T}$ is the EEG signal, and $y \in \{1, \ldots, K\}$ is a label indicating the index of semantic class in the dataset, stored in the form of one-hot encoding. $C$ and $T$ represent the channel dimension and the number of time steps respectively, while $K$ is the number of semantic classes and $N$ is the number of samples. Based on this definition, for each subject, we have datasets $(\mathcal{D}^{img}, \mathcal{D}^{aud}, \mathcal{D}^{tmr})$, which represent image-evoked, audio-evoked, and TMR-related EEG dataset respectively.

The goal of this work is to build a Universal Sleep Decoder (USD) based on the entire dataset $\{(\mathcal{D}_s^{img}, \mathcal{D}_s^{aud}, \mathcal{D}_s^{tmr})\}_{s \in \mathcal{S}}$, with $\mathcal{S}$ the set of subjects, believing that the incorporation of datasets from various subjects, coupled with (resource-rich) awake EEG signals $(\mathcal{X}^{img}, \mathcal{X}^{aud})$ together is beneficial for learning generalizable and discriminative representations of (resource-poor) sleep EEG signals $\mathcal{X}^{tmr}$ (Baltrušaitis et al., 2018; Wang et al., 2022a).

## 3.2 ARCHITECTURE

A general decoder framework commonly consists of a neural encoder $f_{enc}$ and a classification head $f_{cls}$, see Fig.2. The neural encoder $f_{enc}$ maps the neural signal $x$ to the latent feature $z \in \mathbb{R}^F$, with $F$ the latent dimension. And the classification head $f_{cls}$ maps the latent feature $z$ to the label distribution $\hat{y}$. In this work, we implement two kinds of neural encoders, namely CNN-based and Transformer-based neural encoders. Both of them have different configurations in different training settings. More details can be found in Appendix.C.

How semantic classes are represented in the brain during sleep is largely unknown (Türker et al., 2022). We can train the sleep decoder in a supervised manner, similar to what we commonly do when decoding awake neural signals. Specifically, when the neural encoder $f_{enc}$ and the classification head $f_{cls}$ belong to a parameterized family of models such as deep neural network, they can be trained with a classification loss $\mathcal{L}_{cls}(y, \hat{y})$ (e.g. the Cross-Entropy Error),

$$\min_{f_{enc}, f_{cls}} \sum_{x,y} \mathcal{L}_{cls}(y, f_{cls}(f_{enc}(x))). \tag{1}$$

Empirically, we observed that directly applying this supervised decoding approach to sleep data faces several challenges:

1. the limited number of annotated samples within each subject.
2. the noisy signals, and the potentially unreliable labels (e.g. not all audio cues induce the desired cognitive processes (Türker et al., 2022)).

These challenges lead to significant overfitting issues. In comparison, awake datasets $(\mathcal{D}^{img}, \mathcal{D}^{aud})$ are more resource-rich, as we can obtain more samples with reliable annotations easily. As mentioned before, these datasets can be seen as different domains due to the large gap in their neural patterns. To encourage the similarity among features sharing the same semantic class but coming from different domains, we introduce an additional contrastive loss to the training objective:

$$\mathcal{L}_{total} = \mathcal{L}_{cls} + \lambda \mathcal{L}_{contra}, \tag{2}$$

where $\lambda$ is a weighting coefficient to balance the losses. To introduce the label information, we follow the setting of supervised contrastive loss (Khosla et al., 2020). To balance the proportion of sleep data during training, we oversample sleep data for each training batch, ensuring the model encounters both awake and sleep data equally.

Here, we take image-evoked dataset $\mathcal{D}^{img}$ and TMR-related dataset $\mathcal{D}^{tmr}$ for example. For each batch, we draw equal (i.e., $\frac{|\mathcal{B}|}{2}$) sample pairs from these datasets respectively, shuffle them, and then reassemble them into a new dataset. Thus, we have $\mathcal{B} = \{(x_i, y_i)\}_{i=1}^{|\mathcal{B}|}$, while each item $(x_i, y_i)$ is from either $\mathcal{D}^{img}$ or $\mathcal{D}^{tmr}$. Then, each neural signal $x_i$ is mapped to the latent feature $z_i$ through the same neural encoder $f_{enc}$. The contrastive loss is computed by:

$$\mathcal{L}_{contra} = - \sum_{i \in \{1, \ldots, |\mathcal{B}|\}} \frac{1}{|\mathcal{P}(i)|} \sum_{k \in \mathcal{P}(i)} log \frac{e^{\langle z_i, z_k \rangle}}{\sum_{j \in \mathcal{A}(i)} e^{\langle z_i, z_j \rangle}} \tag{3}$$

where $\mathcal{A}(i) = \{1, \ldots, |\mathcal{B}|\} \setminus \{i\}$, $\mathcal{P}(i) = \{k | k \in \mathcal{A}(i), y_k = y_i\}$, and $\langle \cdot, \cdot \rangle$ the inner product.

### 3.3 PRETRAIN-FINETUNE PIPELINE

Our method consists primarily of two stages: the pretraining stage and the finetuning stage. During the pretraining stage (see Fig.2), we leave one subject out, e.g. subject $i$. Then, we use the datasets from the rest subjects to format the training dataset $\mathcal{D}_{train} = \{(\mathcal{D}_s^{img}, \mathcal{D}_s^{aud}, \mathcal{D}_s^{tmr})\}_{s \in \mathcal{S} \setminus \{i\}}$. Following the previous training procedure, we get a pretrained model for that subject. During the finetuning stage, the pretrained model can either be directly evaluated on the sleep data of the test subject, i.e., $\mathcal{D}_i^{tmr}$, or be fine-tuned on part of that dataset before evaluation. Most of the time, the zero-shot setting is preferred because sleep data is usually difficult to collect and the amount of sleep data for each subject is extremely limited. While fine-tuning can lead to improved performance, it also demands more computing resources. We explore both use cases in this work.

As mentioned before, we implement two kinds of neural encoders, and these neural encoders are designed for different purposes. Since the self-similarity operation in the Transformer provides a modeling method that is more adaptive and robust than the convolution operation (Hoyer et al., 2022), the Transformer-based neural encoder is more suitable for learning subject-agnostic features. Consequently, we primarily employ the Transformer-based neural encoder for zero-shot evaluation. To encourage the CNN-based neural encoder to learn subject-agnostic features, we introduce the "Subject Block" into it, and the CNN-based neural encoder is mainly used for fine-tune evaluation.

The "Subject Block" is composed of a $1 \times 1$ convolution layer without activation and a "Subject Layer", which can better leverage inter-subject variability (Défossez et al., 2022; Haxby et al., 2020). Specifically, we learn a matrix $M_s \in \mathbb{R}^{D \times D}$ for each subject $s \in \mathcal{S}$ and apply it after the $1 \times 1$ convolution layer along the channel dimension.

## 4 EXPERIMENTS

In this section, we examine the Universal Sleep Decoder (USD) to validate two research hypotheses:

1. The inclusion of (resource-rich) awake data reduces the overfitting issue caused by the noisy nature of sleep data.

2. Incorporating datasets from various subjects assists the model in acquiring subject-agnostic features, ultimately resulting in improved performance.

### 4.1 DATASETS

As mentioned before, humans are still able to receive and process external sensory stimuli during sleep, even at the semantic level. Therefore, following our designed experiment paradigm (see Fig.1), we can collect neural signals sharing the same semantic neural patterns during both wakefulness and sleep. Besides, the experiment paradigm also provides clear timing and content of memory reactivation during sleep. These two properties together render our dataset a foundational benchmark dataset to validate our research hypotheses. Given the absence of publicly available sleep datasets, we validate our model on the sleep dataset collected by ourselves.

In our dataset, non-invasive EEG recordings were collected during 5 sessions from 52 healthy subjects. Notably, the data for 40 subjects were collected in one laboratory, whereas the data for the remaining 12 subjects were gathered in a different laboratory. Approximately, 12 hours of data were recorded using a 64-channel EEG cap from each subject; see Appendix.A. All subjects share a common set of 15 semantic classes. Before the downstream analysis, we have dropped bad trials before the downstream analysis, see Appendix.B. For each subject, we get 2000 image-evoked EEG signals, 2000 audio-evoked EEG signals, and 1000 TMR-related EEG signals, all with well-balanced labels. Before training, these EEG signals were filtered within the frequency range of 0.1-50Hz, resampled to 100Hz, and subsequently epoched from -0.2s to 0.8s according to the onset of stimuli cue (e.g. image, audio).

### 4.2 RESULTS ON SINGLE-SUBJECT TRAINING SETTING

To validate the first hypothesis, we first evaluate the performance of decoders with awake and sleep datasets respectively. Then, we investigate the potential of transfer from awake dataset to sleep

dataset. Finally, we evaluate the performance of decoders with the integration of awake and sleep datasets. The results for each subject are averaged across 5 seeds.

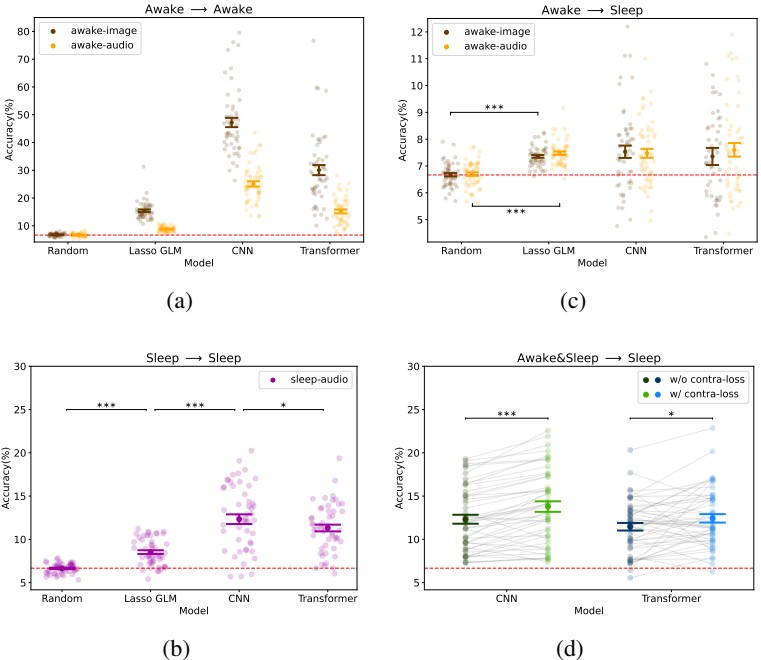

Figure 3: **Results on single-subject training setting. (a).** Performance on awake dataset of decoders trained on awake dataset. We evaluate the performance of different models on image-evoked dataset and audio-evoked dataset respectively. Different points represent the performance of different subjects. Over these points, we plot the mean performance along with the standard error. **(b).** Performance on sleep dataset of decoders trained on sleep dataset. Paired T-tests is performed between different models. **(c).** Performance on sleep dataset of decoders trained on awake dataset. **(d).** Performance on sleep dataset of decoders trained on both awake dataset and sleep dataset. "w/o contra-loss" refers to setting the contrastive loss scale factor $\lambda$ to $0$, while "w/ contra-loss" refers to setting $\lambda$ to $0.5$. The asterisks in figure indicate statistical significance, one asterisk corresponds to a significance level (p-value) below 0.05, two asterisks below 0.01, and three asterisks below 0.001.

### 4.2.1 BASELINE PERFORMANCE OF AWAKE AND SLEEP DATASETS

We implement four different decoding models to evaluate the baseline performance of different datasets (i.e., image-evoked, audio-evoked, and TMR-related datasets) for each subject separately. The "Random" model predicts a uniform distribution over the semantic classes. Logistic GLM follows the standard setup in neuroscience (Liu et al., 2021): we train and evaluate a time-specific classifier for each time point, then take the maximum accuracy among these classifiers. CNN-based model and Transformer-based model follow the model configuration in the single-subject training setting; see Appendix.C for more details.

For each dataset of subject $i$, i.e., $\mathcal{D}_i^{img}$, $\mathcal{D}_i^{aud}$, $\mathcal{D}_i^{tmr}$, we split the dataset into training, validation, testing splits with a size roughly proportional to $80\%$, $10\%$, and $10\%$. We train each model with the training split for 200 epochs, and then evaluate its performance on validation and testing splits. We take the test accuracy according to the maximum validation accuracy as its performance.

Our model with CNN-based neural encoder achieves $12.3\%$ (averaged over 52 subjects) on TMR-related dataset, which is significantly above the random level, see Fig.3(b). This result aligns with prior research findings (Türker et al., 2022) — even during NREM sleep, the human brain still responds to external stimuli. We use this result as the baseline performance in the multi-subject training setting (see Fig.4). Besides, our model achieves $49.5\%$ on image-evoked dataset and $30.2\%$ on audio-evoked dataset; see Fig.3(a).

### 4.2.2 Transfer from awake dataset to sleep dataset

Due to the large gap in neural patterns between awake and sleep neural signals, the transfer from awake neural signals to sleep neural signals remains under exploration. To investigate the potential of transfer, we directly train these decoders on awake datasets (i.e., $\mathcal{D}^{img}$, $\mathcal{D}^{aud}$) respectively, then evaluate the performance with the sleep dataset (i.e., $\mathcal{D}^{tmr}$). We perform paired T-tests between different models, revealing that decoders trained with awake dataset are significantly above the "Random" model (with $p < 0.001$). This finding demonstrates the potential of abundant awake data to enhance sleep decoding.

### 4.2.3 Improve sleep decoding with awake dataset

As we have demonstrated the potential of transfer, a more interesting question is whether we can further improve the decoding performance with additional awake dataset compared to the baseline model trained solely on sleep dataset. Similarly, for each subject, we split the TMR-related dataset $\mathcal{D}^{tmr}$ into training, validation, and testing splits. Then, we introduce additional awake dataset (i.e., $\mathcal{D}^{img}$ and $\mathcal{D}^{aud}$) into the train split. We train each model, either with or without contrastive loss, using the training split for 200 epochs, and then evaluate its performance on validation and testing splits. In our experiment, "w/ contra-loss" commonly refers to setting the contrastive loss scale factor $\lambda$ to 0.5, while "w/o contra-loss" refers to setting $\lambda$ to 0. For further experiments with solely additional $D^{img}$ or $D^{aud}$, see Appendix.D for more details.

In Fig.3(d), the paired T-tests on CNN-based models demonstrate that the model with contrastive loss performs significantly better than that without contrastive loss (with $p < 0.001$). This result validates our first hypothesis — the inclusion of (resource-rich) awake data reduces the overfitting issue caused by the noisy nature of sleep data.

### 4.3 Results on multi-subject training setting

To validate the first hypothesis, we pretrain the proposed Universal Sleep Decoder (USD) following the pipeline in Fig.2, then evaluate its performance in both zero-shot and fine-tune cases.

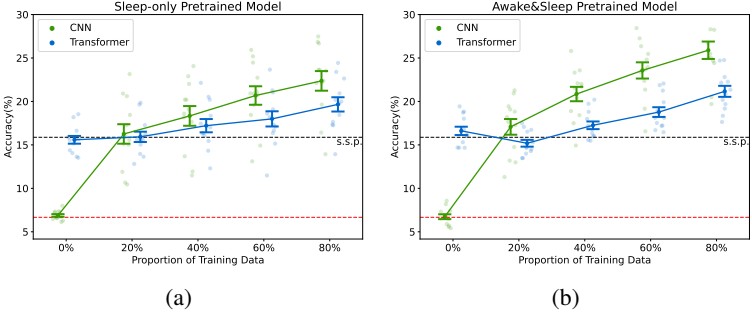

(a)                                             (b)

Figure 4: **Results on multi-subject training setting.(a).** Fine-tune performance of decoders pretrained only on sleep dataset. Different points represent the performance of different subjects. Over these points, we plot the mean performance along with the standard error. "s.s.p." refers to the mean performance of CNN averaged over the corresponding subjects in the single-subject training setting; see Fig.3(b).**(b).** Fine-tune performance of decoders pretrained on awake & sleep dataset. As every sleep data item is paired with one image data item and one audio data item (randomly selected from the awake dataset of the same subject), the amount of finetune data used in awake & sleep pretrained model is three times the amounts of finetune data used in sleep pretrained model.

As mentioned before, the whole dataset comes from two different laboratories. According their sources, we split the set of subjects $\mathcal{S}$ (containing 52 subjects) into two subsets: $\mathcal{S}_1$ (containing 40 subjects), and $\mathcal{S}_2$ (containing 12 subjects). During the pretraining stage, the datasets $\{(\mathcal{D}_s^{img}, \mathcal{D}_s^{aud}, \mathcal{D}_s^{tmr})\}_{s \in \mathcal{S}_1}$ from $\mathcal{S}_1$ are used for the supervised pretraining of USD. In total, there are approximately $80,000$ image-evoked data, $80,000$ audio-evoked data, and $40,000$ TMR-related

data across 40 subjects for supervised pretraining USD. Then, the datasets $(\mathcal{D}_i^{img}, \mathcal{D}_i^{aud}, \mathcal{D}_i^{tmr})$ of subject $i \in \mathcal{S}_2$ are used for latter evaluation.

In the single-subject training setting, the inclusion of (resource-rich) awake data leads to improved performance; see Fig.3(d). Here, we pretrain USD separately on sleep-only datasets (i.e., $\{\mathcal{D}_s^{tmr}\}_{s \in \mathcal{S}_1}$) and the whole datasets (i.e., $\{(\mathcal{D}_s^{img}, \mathcal{D}_s^{aud}, \mathcal{D}_s^{tmr})\}_{s \in \mathcal{S}_1}$), to further investigate that hypothesis in the multi-subject training setting. Due to the limited computing resources, we only evaluate these experiments on the 12 subjects from $\mathcal{S}_2$. Since the data of $\mathcal{S}_1$ and $\mathcal{S}_2$ comes from two different laboratories, the experiments can better validate the generalization ability of our model. The results for each subject are averaged across 5 seeds.

### 4.3.1 ZERO-SHOT RESULT ON SLEEP DECODING

After the pretraining stage, we directly apply the pretrained model for sleep decoding on the held-out subject, one that the model has not previously encountered. We investigate the zero-shot ability of USD with CNN-based and Transformer-based neural encoders separately; see Fig.4(b). As mentioned before, we introduce "Subject Block" into the CNN-based neural encoder, thus encouraging the following feature extractor to learn subject-agnostic features. Since the "Subject Block" has never seen that subject before, the "Subject Block" cannot map the data of that subject to the subject-agnostic space, which leads to random level zero-shot ability (see Fig.4(a)).

In comparison, USD with the Transformer-based neural encoder attains $15.58\%$ zero-shot accuracy across 12 subjects, which is quite impressive, considering it's comparable to the baseline model trained in the single-subject training setting. Furthermore, by incorporating additional awake dataset during pretraining, the Transformer-based model achieves higher accuracy, $16.61\%$.

### 4.3.2 FINE-TUNE RESULT ON SLEEP DECODING

After the pretraining stage, we can also fine-tune the pretrained model with some sleep data of that subject. We investigate the fine-tune performance of USD with CNN-based and Transformer-based neural encoders separately; see Fig.4(b). When pretrained with sleep-only dataset, USD with the CNN-based neural encoder achieves $22.4\%$ with 80% TMR-related dataset of that subject, surpassing the baseline model trained in the single-subject training setting. This result lends support to the second hypothesis — incorporating datasets from various subjects assists the model in acquiring subject-agnostic features, ultimately resulting in improved performance. With additional awake dataset during the pretrain stage, the CNN-based USD achieves higher accuracy, $25.9\%$, which validates the first hypothesis in the multi-subject training setting. Besides, as the amount of fine-tuning data increases, the performance of the USD gradually improves.

In comparison, USD with the Transformer-based neural encoder achieves $19.7\%$ and $21.2\%$ respectively, with 80% TMR-related dataset of that subject. However, the trend of its performance with the amount of fine-tuning data is slightly different from that of the CNN-based one. The Transformer-based model with 20% TMR-related dataset performs slightly worse than the zero-shot baseline, which is normal as the restricted amount of fine-tuning data can hinder the knowledge acquired by the feature extractor (Shen et al., 2021).

## 5 CONCLUSION & LIMITATIONS

In this study, we introduce the Universal Sleep Decoder (USD), a model designed to align awake and sleep neural representations across subjects. This alignment creates representations that are both subject-agnostic and domain-agnostic, significantly enhancing the accuracy and data-efficiency of sleep decoding, even on unseen subjects. This advancement is crucial in neuroscience, a field often operating in a small data regime. The acquisition of sleep data is a substantial investment; therefore, any reduction in the data required for decoding has a profound impact. Furthermore, due to the similarities in high-temporal-resolution brain recordings, our model can be effectively adapted to MEG or sEEG. In the future, our model holds the potential for real-time sleep decoding, offering a robust tool for manipulating memory reactivation during sleep in real-time. One limitation of our study is that we cannot distinguish the specific memory content from its corresponding high-level semantics. In this study, we view them as the same thing.

ETHICS STATEMENT

The research that has been documented adheres to the ethical guidelines outlined by the ICLR. The data acquisition and the follow-up experiments were approved by the local ethics community in *anonymous organization* (ethics review ID: *ICBIR_A_0204_005*). Every participant signed an informed consent form, acknowledging their rights. Participants were compensated with cash.

REPRODUCIBILITY STATEMENT

Code to train models and reproduce the results was submitted as part of the supplementary materials.

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

# A EXPERIMENT DESIGN

As demonstrated in Fig.1, our experiment paradigm contained three stages: pre-sleep function localizer stage, overnight sleep target memory reactivation stage, and post-sleep function localizer stage. First, we selected 15 semantic classes: alarm, apple, ball, book, box, chair, kiwi, microphone, motorcycle, pepper, sheep, shoes, strawberry, tomato, watch. Then, we used the corresponding pictures and sounds to form the image-set (i.e., containing 15 different pictures) and the audio-set (i.e., containing 15 different sounds). It's worth noting that the sound corresponding to the picture can easily trigger the recall of the associated picture, e.g., the picture of "sheep" is paired with the sound of the "sheep" instead of other unrelated sounds.

## A.1 PRE&POST-SLEEP FUNCTION LOCALIZER STAGE

In the pre-sleep function localizer stage, the subject was instructed to perform two tasks: the image-audio task and the audio-image task. In image-audio task, the subject was presented with 600 trials, with each trial containing one picture and one sound, in image-audio order. We randomly selected one picture and one sound from image-set and audio-set respectively, to form an image-audio pair. In each image-audio pair, the picture and the sound may be inconsistent. In total, we had 210 inconsistent trials and 390 consistent trials. The "consistent trial" means the sound and the picture in that trial belong to the same stimulus pair, and the "inconsistent trial" means the sound and the picture in that trial don't belong to the same stimulus pair. The purpose of this design is solely to confirm that participants had formed correct and stable stimulus pair memories.

For each trial, the subject was presented with the following items in sequence:

- a blank screen for 0.6s-0.9s (i.e., inter-trial interval),

- a cross centered at the screen for 0.3s,

- an image cue of that image-audio pair for 0.8s,

- a blank screen for 0.3s,

- an audio cue of that image-audio pair for 0.5s,

- a blank screen for 0.3s.

Then, the subject was required to decide whether the presented image cue and audio cue are matched. Finally, we presented the behavior feedback for 0.2s. In total, image-audio task took about 48 minutes.

The setup of audio-image task was similar to that of image-audio task. In audio-image task, the only difference from image-audio task was that we presented image-audio pairs in audio-image order.

In the post-sleep function localizer stage, the subject was required to execute the same tasks in the pre-sleep function localizer stage.

## A.2 OVERNIGHT SLEEP TARGET MEMORY REACTIVATION STAGE

In the overnight sleep target memory reactivation stage, we utilized a closed-loop stimulation system that allows for real-time, automatic sleep staging. As the subject reached the N2/3 stage of NREM sleep, the subject was presented with audio cues (which are the same as those during wakefulness, each lasting 0.5s), randomly selected from audio-set, every 4-6 seconds. This approach provided clear timing and content of memory reactivation during sleep, aiding in the training of a neural decoder on these cued sleep intervals.

## A.3 DATASET SUMMARY

In total, we collect datasets from 52 subjects (40 subjects from one laboratory, 12 subjects from another laboratory). This operation is to better demonstrate the generalization ability of our model.

# B    PREPROCESSING

EEG signals are filtered within the frequency range of 0.1-50Hz. Line noise at 50Hz and its harmonics are removed. Each channel is re-referenced by subtracting out the mean signal over all EEG channels (average re-referencing). This decreases the effect caused by online-reference channel selection (Yao, 2001). Then we apply Independent Component Analysis (ICA) and remove artifact components automatically (Li et al., 2022a). The preprocessed EEG signals are further resampled to 100Hz. Finally, we epoch the EEG signals from -0.2s to 0.8s according to the onset of the stimuli cue (e.g. image, audio), and drop noisy epochs automatically. All of the above operations are done with MNE-Python (Gramfort et al., 2013).

The preprocess pipeline of sleep data contains one more step, i.e., sleep staging, at the very beginning. To this aim, we follow the standard pipeline of YASA-Python (Vallat & Walker, 2021). According to the recommendation of the algorithm, we provide a central electrode $C3$ referenced to the mastoid $M2$ (i.e., $C3 - M2$) as the EEG channel, the right EOG channel $REOG$ referenced to the mastoid $M2$ (i.e., $REOG - M2$) as the EOG channel, and the right EMG channel $REMG$ referenced to the reference EMG channel $EMGREF$ (i.e., $REMG - EMGREF$) as the EMG channel. After the sleep staging, we truncate EEG signals according to NREM 2/3 sleep stage, and splice them together in sequence to form the TMR-related EEG signals of that subject.

# C    MODEL PARAMETERS

## C.1    MODEL PARAMETERS IN SINGLE-SUBJECT TRAINING SETTING

The USD model consists of two parts, the neural encoder and the classification head. The neural encoder is implemented with either CNN or Transformer. The classification head contains one Linear layer ($d_h = 128$) without activation, and one following Linear layer ($d_h = 15$) with softmax activation. The configuration of CNN-based model is listed in Table.1 and the configuration of Transformer-based model is listed in Table.2.

### C.1.1    CNN-BASED MODEL PARAMETERS

The CNN-based neural encoder disables the "Subject Block". The "Encoder Block" starts with convolution layers ($N = 2$). A MaxPool1D layer along the time-axis with pooling kernel as 2 is followed. In the following "Feature Block", all feature maps are flattened through a Flatten layer. Finally, one Dropout layer ($p_{dropout} = 0.5$) and one BatchNormlization layer are added at the end.

Table 1: The configuration of CNN-based model in single-subject training setting

| blocks | layers | # of filters | kernel size | dilation rate |
|---|---|---|---|---|
| | Conv1D | 256 | 9 | 1 |
| Encoder | Conv1D | 128 | 11 | 2 |
| | MaxPool1D | 2 | - | - |
| | Flatten | - | - | - |
| Feature | Dropout | rate = 0.5 | - | - |
| | BatchNorm | - | - | - |
| Contrastive | Linear | 128 | - | - |
| Classification | Linear | 128 | - | - |
| | Linear | 15 | - | - |

To calculate the contrastive loss, we further map the latent vector $z \in \mathbb{R}^{1 \times F}$ through a Linear layer ($d_h = 128$) to get the hidden vector $h \in \mathbb{R}^{1 \times d_h}$. We then employ Eq.(3) to calculate the corresponding contrastive loss.

The model is trained up to 200 epochs by default using Adam (Kingma & Ba, 2014) with a learning rate of $3 \cdot 10^{-4}$ and a batch size of 64 on one NVIDIA V100 GPU with 32GB of memory.

### C.1.2 TRANSFORMER-BASED MODEL PARAMETERS

In the Transformer-based neural encoder, all attention blocks ($N = 2$) are set with the following parameters: $d_h = 256$, $n_{head} = 8$, and $d_{head} = 128$. The dropout ratio of the second Feed-Forward layer is set to $0.3$ for each attention block. Finally, all feature maps are flattened through a Flatten layer. Furthermore, the configuration of the classification head remains the same with that of CNN-based model.

Table 2: The configuration of Transformer-based model in single-subject training setting

| blocks | layers | $d_{hidden}$[1] | $n_{head}$[2] | $d_{head}$[3] | other parameters |
|---|---|---|---|---|---|
| Encoder | Attention | 256 | 8 | 128 | $dropout_{ff} = [0., 0.3]$ |
|  | Attention | 256 | 8 | 128 | $dropout_{ff}$[4]$= [0., 0.3]$ |
| Feature | Flatten | - | - | - | - |
| Contrastive | Linear | 128 | - | - | - |
| Classification | Linear | 128 | - | - | - |
|  | Linear | 15 | - | - | - |

[1] $d_{hidden}$ refers to the dimensions of the hidden layer in Feed Forward Network.

[2] $n_{head}$ refers to the number of attention heads.

[3] $d_{head}$ refers to the dimensions of attention head.

[4] $dropout_{ff}$ refers to the dropout probability of the hidden layer in Feed Forward Network.

The model is trained up to 200 epochs by default using Adam with a learning rate of $3 \cdot 10^{-4}$ and a batch size of 64 on one NVIDIA V100 GPU with 32GB of memory.

### C.2 MODEL PARAMETERS IN MULTI-SUBJECT TRAINING SETTING

In the multi-subject training setting, we also implemented CNN-based and Transformer-based neural encoders. Considering the large amount of data from 40 subjects, we appropriately scaled up the model size and adjusted the corresponding parameters. Similarly, due to the different amount of data when training on the sleep-only dataset and the awake&sleep dataset, specific model parameters also undergo adjustments. The parameters of the classification head are similar to those in the single-subject training setting. The classification head further includes one Flatten layer and Dropout layer ($p_{dropout} = 0.5$) at the very beginning. Accordingly, the latent space before the classification head is 2D, containing additional time information. This design follows the multi-subject training setting in Défossez et al. (2022), which generally leads to better performance in our experiments. The configuration of CNN-based model in multi-subject training is listed in Table.3 and the configuration of Transformer-based model in multi-subject training is listed in Table.4.

### C.2.1 CNN-BASED MODEL PARAMETERS

Since the model needs to handle EEG data from different subjects, the CNN-based neural encoder enables the "Subject Block", which can better leverage inter-subject variability with the unique identification numbers of subjects. The configuration of the "Subject Block" is listed in Table.3. Following the Conv1D layer, we further add one BatchNormalization layer, one Dropout layer ($p_{dropout} = 0.3$), and two Linear layers with different parameters. The final component of the subject block is the subject layer, serving as its core. The Subject Layer maps different subject data to a shared latent space according to their own data index. The idea of having a layer that transforms input data from various subjects into a common feature space, enabling the model to leverage information from different subjects while still allowing for subject-specific variations. This type of approach can enhance the model's ability to generalize patterns across a diverse set of subjects.

The configuration of Encoder Block is listed in Table.3 and starts with convolution layers ($N = 5$). A MaxPool1D layer along the time-axis with pooling kernel as $4$ is followed. Then, we add one

Dropout layer ($p_{dropout} = 0.5$), and added one Linear layer ($d_h = 160$ for the sleep-only model; $d_h = 256$ for the awake-sleep model), without activation.

Additionally, for datasets encompassing both wakefulness and sleep states, we employ two contrastive blocks to compute the loss: one between audio-evoked and image-evoked EEG data during wakefulness, and the other between audio-evoked and TMR-related EEG data. In the contrastive block, we map the latent vector $z \in \mathbb{R}^{C \times F}$ through a Linear layer ($d_h = 128$) to get the hidden vector $h \in \mathbb{R}^{C \times d_h}$. We then employ Eq.(3) to calculate the corresponding contrastive loss.

It's crucial to reiterate that for the entire model, only the convolution layers are shared. Conversely, the remaining components, (i.e., the "Subject Block", and the classification head) have different instantiates for different data domains (i.e., $\mathcal{R}^{img}, \mathcal{R}^{aud}, \mathcal{R}^{tmr}$).

For the pretraining of both models, each model is trained up to 500 epochs by default using Adam with a learning rate of $5 \cdot 10^{-5}$. For the fine-tuning of these models, we freeze the Conv1D layer in the "Subject Block" and the convolution layers in the neural encoder. Then, each model is fine-tuned up to 200 epochs by default using Adam with a learning rate of $1 \cdot 10^{-5}$.

The pretraining and fine-tuning of both models were done with a batch size of 256 on one NVIDIA V100 GPU with 32GB of memory.

Table 3: The configuration of CNN-based model in multi-subject training setting

| blocks | layers | # of filters (s./a.&s.)[1] | | | kernel size | dilation rate |
|---|---|---|---|---|---|---|
| Subject | Conv1D | 256 | / | 512 | 1 | 1 |
| | Dropout | rate = 0.3 | | | - | - |
| | BatchNorm | - | | | - | - |
| | Linear | 256 | / | 512 | - | - |
| | Linear | 128 | / | 256 | - | - |
| | Subject Layer | - | | | - | - |
| Encoder | Conv1D | 128 | / | 128 | 5 | 1 |
| | Conv1D | 256 | / | 256 | 7 | 1 |
| | Conv1D | 512 | / | 512 | 11 | 2 |
| | Conv1D | 256 | / | 256 | 9 | 2 |
| | Conv1D | 64 | / | 128 | 3 | 2 |
| | MaxPool1D | 4 | | | - | - |
| | Dropout | rate = 0.5 | | | - | - |
| | BatchNorm | - | | | - | - |
| Feature | Linear | 160 | / | 256 | - | - |
| Contrastive | Linear | ✘[2] | / | 128 | - | - |
| | Flatten | ✘ | / | ✔[3] | - | - |
| Classification | Flatten | - | | | - | - |
| | Linear | 128 | / | 128 | - | - |
| | Linear | 15 | / | 15 | - | - |

[1] "s." represents the sleep-only pretrained model, and "a.&s." represents the awake&sleep pretrained model.

[2] ✘ indicates the corresponding layer was not used during pretraining and fine-tuning.

[3] ✔ indicates the corresponding layer was used during pretraining and fine-tuning.

### C.2.2    TRANSFORMER-BASED MODEL PARAMETERS

In the Transformer-based model, we add the "Channel Block" at every beginning, which has a similar structure to the "Subject Block" in the CNN-based model, but without the "Subject Layer".

Similarly, the sleep-only model and the awake-sleep model have different configurations of model parameters. For the sleep-only model, all attention blocks ($N = 8$) are set with the following

parameters: $d_h = 256$, $n_{head} = 8$, and $d_{head} = 128$. For the awake-sleep model, all attention blocks ($N = 10$) are set with the following parameters: $d_h = 512$, $n_{head} = 10$, and $d_{head} = 256$. For both models, the dropout ratio of the second Feed-Forward layer is set to $0.5$ for each attention block. Finally, we add one Linear layer ($d_h = 80$ for the sleep-only model; $d_h = 256$ for the awake-sleep model), without activation. The configuration of the contrastive block is similar to that in the CNN-based model, with slight modification ($d_h = 64$).

It's crucial to reiterate that for the entire model, only the "Attention Block"s are shared. Conversely, the remaining components, (i.e., the "Channel Block", the embedding layer before the "Attention Block"s, and the classification head) have different instantiates for different data domains (i.e., $\mathcal{R}^{img}, \mathcal{R}^{aud}, \mathcal{R}^{tmr}$).

For the pretraining of both models, each model is trained up to 500 epochs by default using Adam with a learning rate of $5 \cdot 10^{-5}$. For the fine-tuning of these models, all parts of each model are unlocked to be trainable. Then, each model is fine-tuned up to 200 epochs by default using Adam with a learning rate of $1 \cdot 10^{-5}$.

The pretraining and fine-tuning of both models were done with a batch size of 256 on one NVIDIA V100 GPU with 32GB of memory.

Table 4: The configuration of Transformer-based model in multi-subject training setting

| blocks | layers | $d_{hidden}$[1] (s./a.&s.)[2] | | | $n_{head}$[3] | $d_{head}$[4] | other parameters |
|---|---|---|---|---|---|---|---|
| Channel | Conv1D | 256 | / | 512 | - | - | kernel size = 1 |
| | Dropout | | rate = 0.3 | | - | - | - |
| | BatchNorm | | - | | - | - | - |
| | Linear | 256 | / | 512 | - | - | - |
| | Linear | 128 | / | 256 | - | - | - |
| Encoder | Attention ×8 | 256 | / | 512 | 8 / 10 | 128 / 256 | $dropout_{ff}$[5]$= [0., 0.5]$ |
| | Attention ×2 | ✘[6] | / | 512 | ✘ / 10 | ✘ / 256 | $dropout_{ff} = [0., 0.5]$ |
| Feature | Linear | 80 | / | 256 | - | - | - |
| Contrastive | Linear | ✘ | / | 64 | - | - | - |
| | Flatten | ✘ | / | ✔[7] | - | - | - |
| Classification | Flatten | | - | | - | - | - |
| | Linear | 128 | / | 128 | - | - | - |
| | Linear | 15 | / | 15 | - | - | - |

[1] $d_{hidden}$ refers to the dimensions of the hidden layer in Feed Forward Network.

[2] "s." represents the sleep-only pretrained model, and "a.&s." represents the awake&sleep pretrained model.

[3] $n_{head}$ refers to the number of attention heads.

[4] $d_{head}$ refers to the dimensions of attention head.

[5] $dropout_{ff}$ refers to the dropout probability of the hidden layer in Feed Forward Network.

[6] ✘ indicates the corresponding layer was not used during pretraining and fine-tuning.

[7] ✔ indicates the corresponding layer was used during pretraining and fine-tuning.

# D EXTRA RESULTS

## D.1 EXTRA RESULTS OF SLEEP ANALYSIS

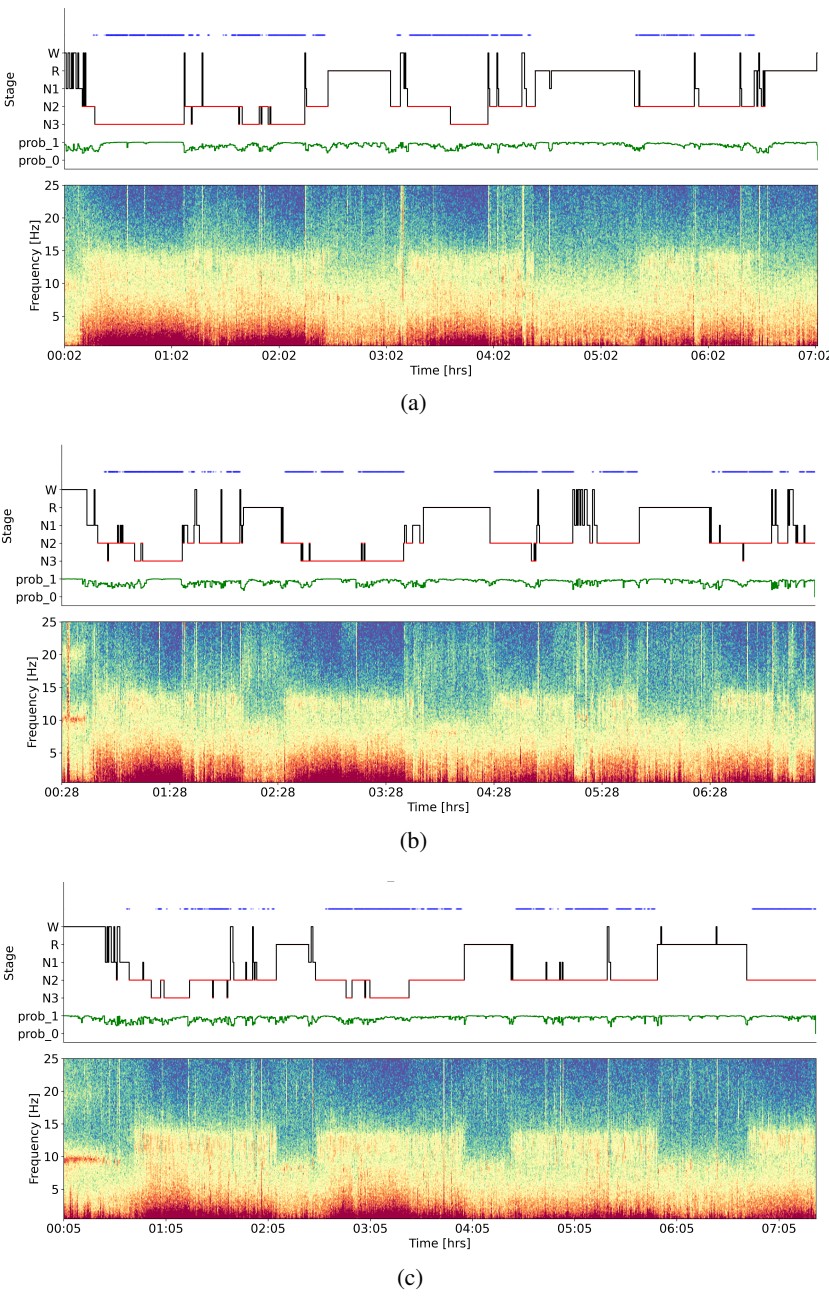

Figure 5: **The illustration of sleep staging and TMR-related stimulus. (a).** Sleep staging and spectrogram of preprocessed EEG sleep data corresponding to the first demo subject. Each blue dot above represents each sound stimulus during sleep. The N2/3 sleep stage (red line) is automatically identified based on YASA toolbox. **(b).** The second demo subject. **(c).** The third demo subject.

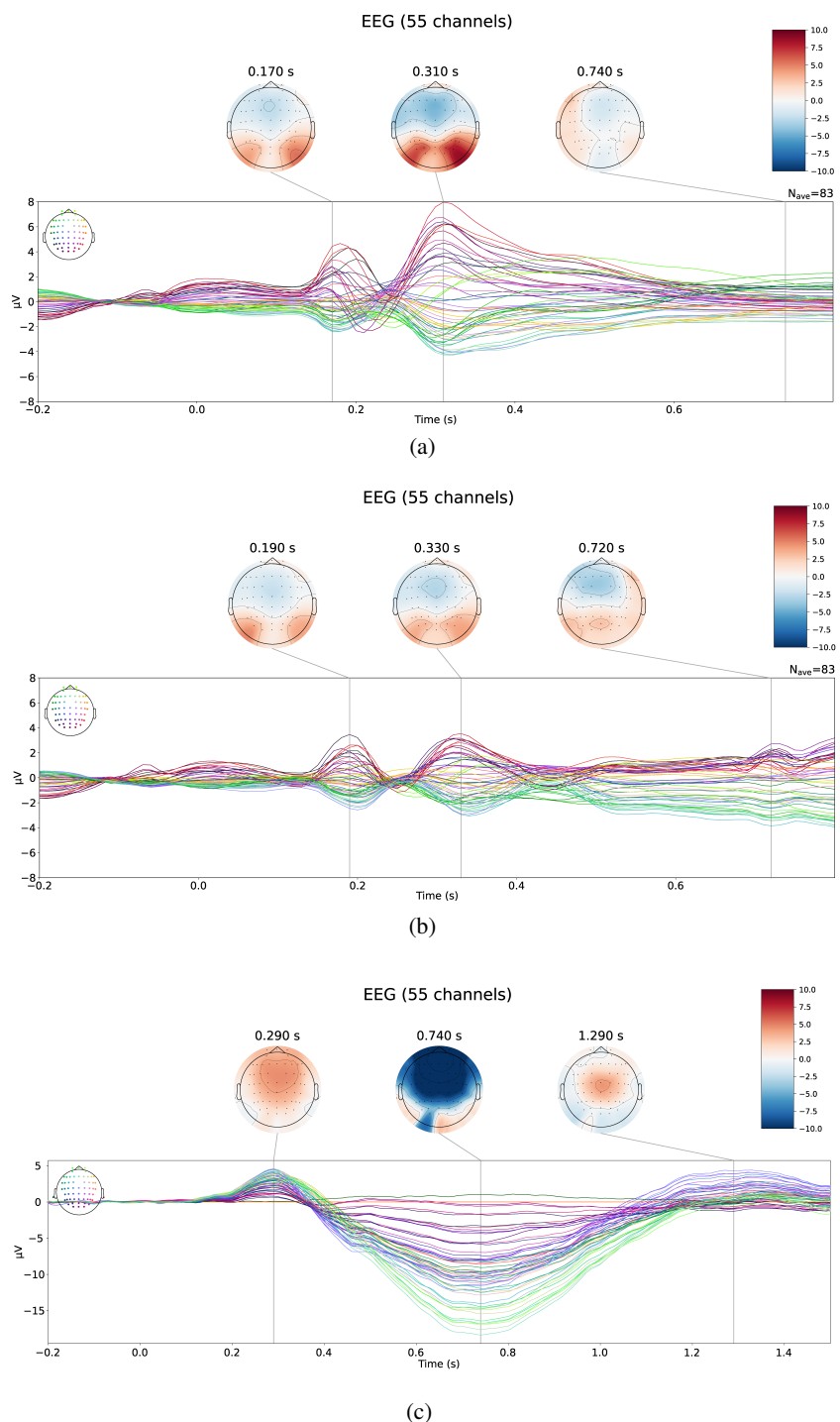

Figure 6: **The event-related potentials (EPRs) of different kind of stimulus. (a).** The average ERPs and the topographic maps evoked by image stimuli during wakefulness (n=52). Due to the experimental design, with a minimum inter-stimulus interval of only 1.0 second, ERPs were computed within a narrow window from 0.2 seconds before stimulus onset to 0.8 seconds after stimulus onset. **(b).** The average ERPs and the topographic maps evoked by audio stimuli during wakefulness (n=52). **(c).** The average ERPs and the topographic maps evoked by TMR-related audio stimuli during N2/3 sleep (n=52). And the minimum inter-stimulus interval during sleep are about 4.0 seconds, so ERPs were computed within a narrow window from 0.2 seconds before stimulus onset to 1.5 seconds after stimulus onset.

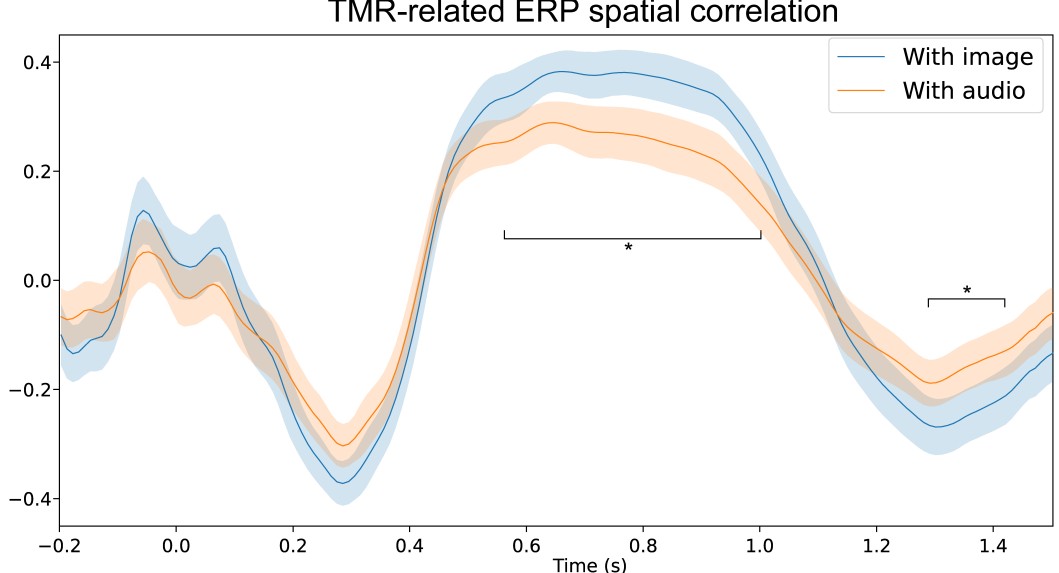

Figure 7: **The spatial correlation between TMR-related ERPs and image/audio ERPs.** Firstly, the time point with the maximum variability in the averaged ERPs for image and audio across all participants was identified. The spatial distributions of ERPs at these two time points for image and audio were considered as the templates for each participant. Subsequently, the spatial correlation between the ERPs at each time point related to TMR and the two templates for each participant was computed. Paired t-tests were then conducted on the two sets of correlation coefficients at each time point, and significant differences were annotated. $^*p < 0.05$.

## D.2 EXTRA RESULTS IN THE SINGLE-SUBJECT TRAINING SETTING

Table 5: The results (mean±ste;%) of different models on awake→awake, sleep→sleep and awake→sleep experiments in single-subject training setting. For each model-experiment pair, we report the mean accuracy and the standard error across subjects, which are averaged over 5 seeds.

|  | image → image | audio → audio | sleep → sleep | image → sleep | audio → sleep |
|---|---|---|---|---|---|
| Lasso GLM | $15.43 \pm 0.45$ | $8.80 \pm 0.13$ | $8.53 \pm 0.21$ | $7.36 \pm 0.05$ | $7.48 \pm 0.06$ |
| CNN | $47.20 \pm 1.67$ | $25.09 \pm 0.95$ | $12.33 \pm 0.56$ | $7.53 \pm 0.23$ | $7.47 \pm 0.17$ |
| Transformer | $30.05 \pm 1.81$ | $15.16 \pm 0.70$ | $11.31 \pm 0.39$ | $7.36 \pm 0.32$ | $7.60 \pm 0.25$ |

Table 6: The results (mean±ste;%) of different models on awake→sleep experiments in single-subject training setting. All results are averaged over 5 seeds. The best results for each experiment have been marked in bold.

|  | image&sleep → sleep | audio&sleep → sleep | awake&sleep → sleep |
|---|---|---|---|
| Lasso GLM | $8.27 \pm 0.20$ | $8.33 \pm 0.20$ | $8.29 \pm 0.15$ |
| CNN ($\lambda = 0$) | $12.39 \pm 0.52$ | $12.26 \pm 0.55$ | $12.33 \pm 0.53$ |
| +contrastive ($\lambda = 0.5$) | $\mathbf{13.67 \pm 0.62}$ | $\mathbf{13.56 \pm 0.60}$ | $\mathbf{13.79 \pm 0.61}$ |
| Transformer ($\lambda = 0$) | $11.25 \pm 0.51$ | $11.29 \pm 0.50$ | $11.37 \pm 0.49$ |
| +contrastive ($\lambda = 0.5$) | $12.21 \pm 0.50$ | $12.15 \pm 0.47$ | $12.42 \pm 0.49$ |

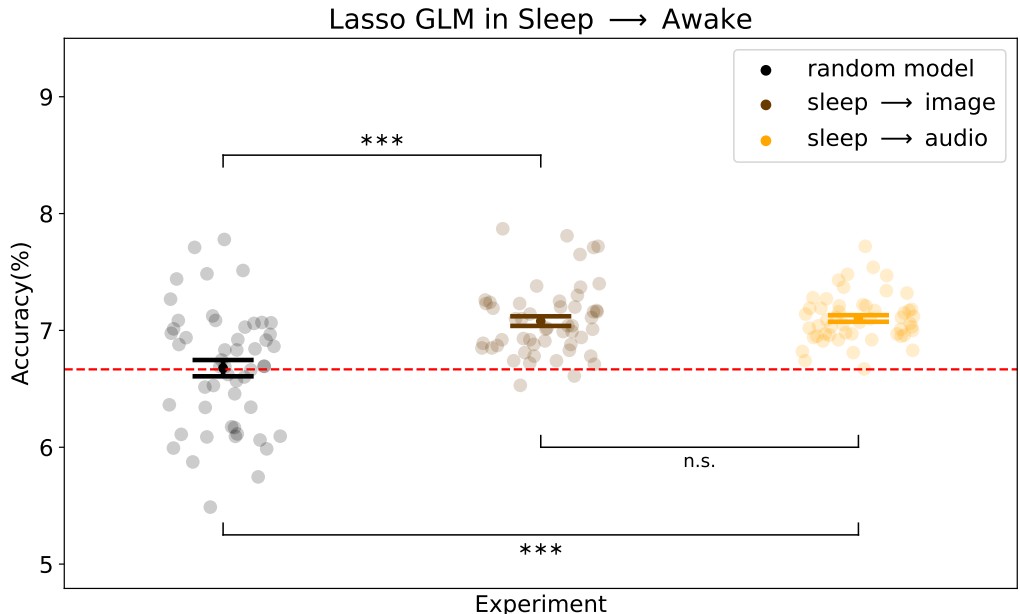

Figure 8: **Results (mean±ste;%) of Lasso GLM on sleep→awake experiments in single-subject training setting.** The sleep decoding performance of Lasso GLM trained either with or without awake dataset. All results are averaged over 30 seeds. When incorporating with awake dataset, the number of samples from these two datasets is balanced.

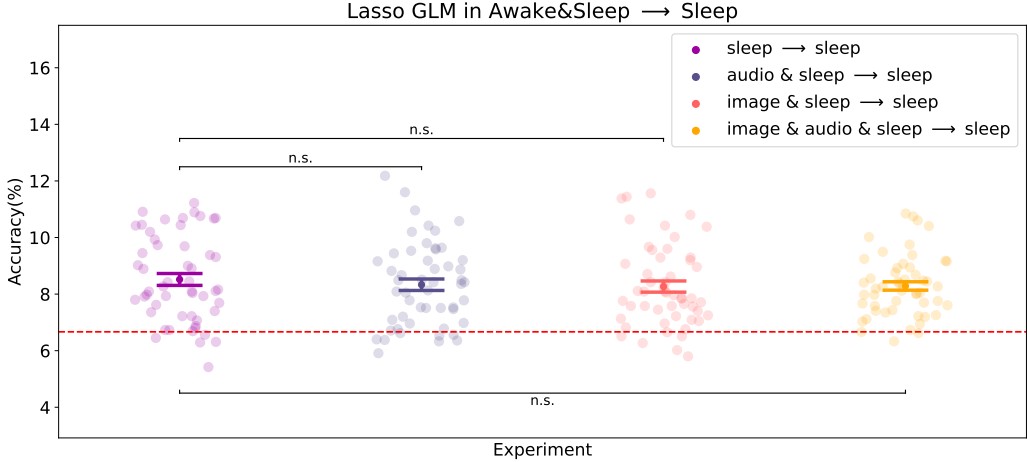

Figure 9: **Results (mean±ste;%) of Lasso GLM on awake&sleep→sleep in single-subject training setting.** The sleep decoding performance of Lasso GLM trained either with or without awake dataset. All results are averaged over 30 seeds. When incorporating with awake dataset, the number of samples from these two datasets is balanced.

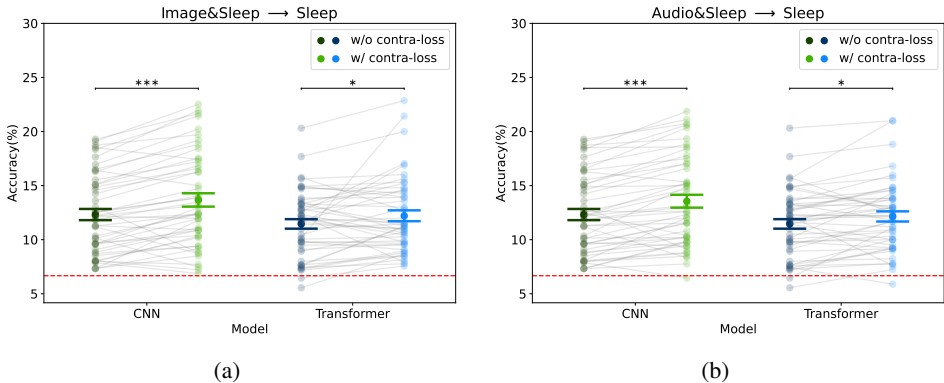

Figure 10: **Extra results (mean±ste;%) of CNN/Transformer on awake&sleep→sleep experiments in single-subject training setting. (a).** Performance on sleep dataset of decoders trained on both awake-image dataset and sleep dataset. "w/o contra-loss" refers to setting the contrastive loss scale factor $\lambda$ to 0, while "w/ contra-loss" refers to setting $\lambda$ to 0.5. **(b).** Performance on sleep dataset of decoders trained on both awake-audio dataset and sleep dataset. All results are averaged over 5 seeds.

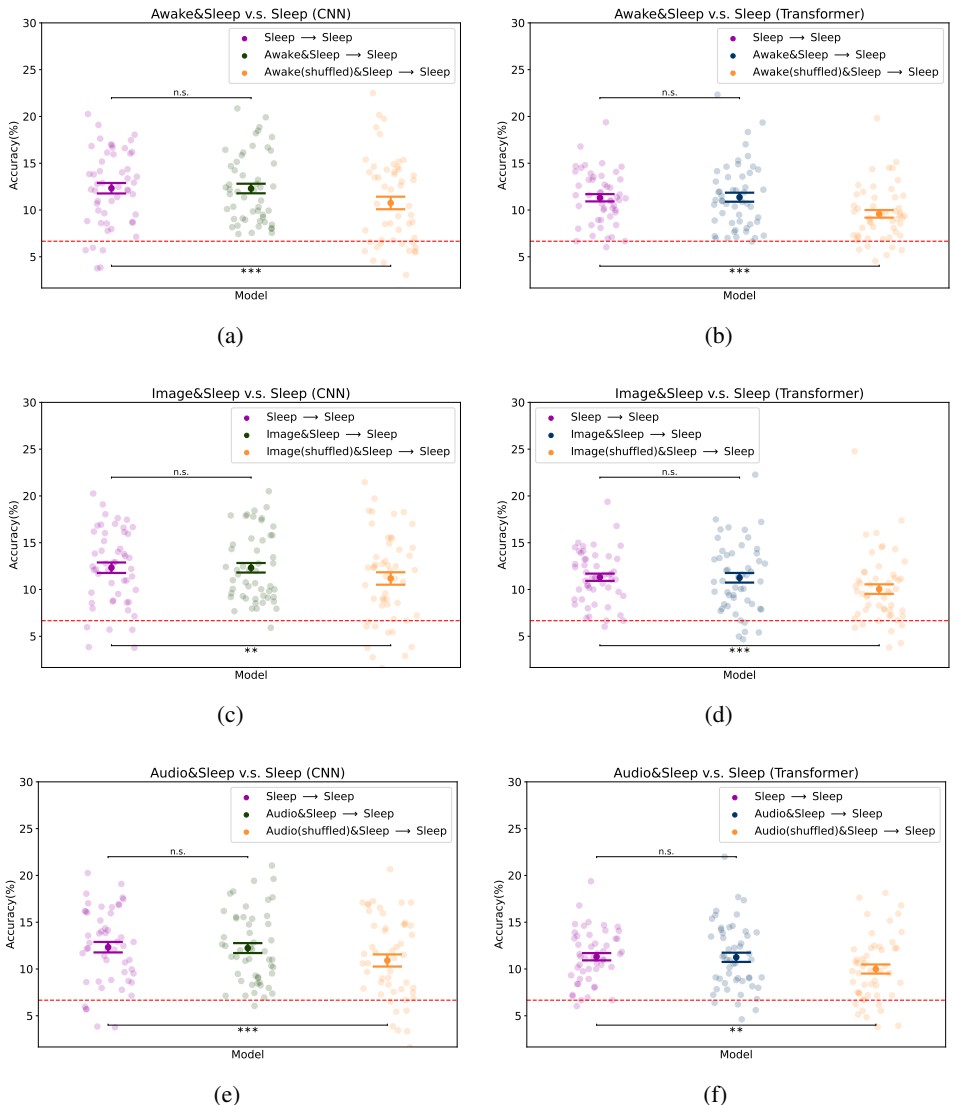

Figure 11: **The effect of additional awake dataset in single-subject training setting.** The sleep decoding performance (mean±ste;%) of each model and each experiment is reported. **(a).** The sleep decoding performance of CNN on awake&sleep→sleep experiment. "Sleep→Sleep" refers to training with only the sleep dataset. "Awake&Sleep→Sleep" refers to training with additional awake dataset. "Awake(shuffled)&Sleep" refers to training with additional label-shuffled awake dataset. **(b).** The sleep decoding performance of Transformer on awake&sleep→sleep experiment. **(c).** The sleep decoding performance of CNN on image&sleep→sleep experiment. **(d).** The sleep decoding performance of Transformer on image&sleep→sleep experiment. **(e).** The sleep decoding performance of CNN on audio&sleep→sleep experiment. **(f).** The sleep decoding performance of Transformer on audio&sleep→sleep experiment. All results are averaged over 5 seeds.

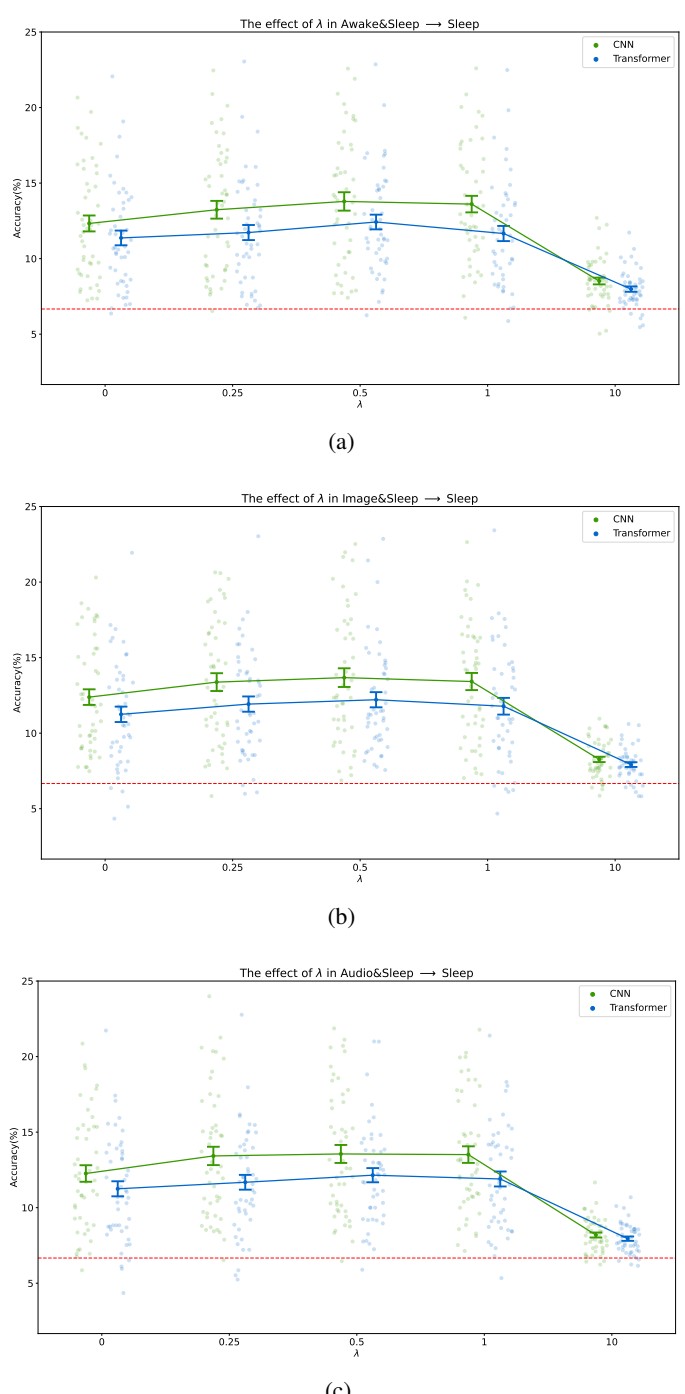

(a)

(b)

(c)

Figure 12: **The effect of** $\lambda$ **on awake&sleep→sleep experiments in single-subject training setting.** The sleep decoding performance (mean±ste;%) of each model and each experiment is reported. **(a).** The effect of $\lambda$ on awake&sleep→sleep experiment. **(b).** The effect of $\lambda$ on image&sleep→sleep experiment. **(c).** The effect of $\lambda$ on audio&sleep→sleep experiment. All results are averaged over 5 seeds.

## D.3 Extra results in the multi-subject training setting

Table 7: The results of finetune performance (mean±std;%) in multi-subject training setting. The performance under subject id is the baseline performance of CNN in the single-subject training setting; see Fig.3(b).

| Subject | Pretrain Model | 0% | 20% | 40% | 60% | 80% |
|---|---|---|---|---|---|---|
| subject 1 (16.2 ± 1.2) | CNN w/o awake | 6.4 ± 0.1 | 23.2 ± 0.4 | 24.1 ± 0.3 | 25.9 ± 0.7 | 26.9 ± 0.4 |
| | CNN w/ awake | 6.9 ± 0.1 | 19.4 ± 0.5 | 23.6 ± 0.5 | 25.5 ± 0.4 | 28.3 ± 0.7 |
| | Transformer w/o awake | 18.2 ± 0.4 | 19.6 ± 0.7 | 20.5 ± 0.6 | 21.0 ± 0.7 | 22.6 ± 0.7 |
| | Transformer w/ awake | 18.7 ± 0.5 | 17.5 ± 0.5 | 19.7 ± 0.3 | 21.9 ± 0.6 | 24.8 ± 0.5 |
| subject 2 (16.7 ± 1.5) | CNN w/o awake | 7.3 ± 0.2 | 20.9 ± 0.7 | 22.0 ± 0.9 | 22.7 ± 0.7 | 23.6 ± 0.6 |
| | CNN w/ awake | 5.9 ± 0.1 | 18.2 ± 0.7 | 22.0 ± 0.5 | 28.4 ± 0.9 | 31.8 ± 1.1 |
| | Transformer w/o awake | 18.5 ± 0.4 | 17.3 ± 0.4 | 18.4 ± 0.6 | 19.1 ± 0.7 | 20.2 ± 0.9 |
| | Transformer w/ awake | 18.8 ± 0.5 | 17.1 ± 0.4 | 20.2 ± 0.7 | 22.1 ± 0.7 | 24.7 ± 0.6 |
| subject 3 (16.3 ± 0.9) | CNN w/o awake | 6.2 ± 0.1 | 18.5 ± 0.7 | 20.2 ± 0.9 | 21.2 ± 0.9 | 22.1 ± 0.8 |
| | CNN w/ awake | 5.7 ± 0.2 | 13.2 ± 0.5 | 18.9 ± 0.4 | 21.4 ± 0.7 | 25.0 ± 0.7 |
| | Transformer w/o awake | 15.6 ± 0.7 | 15.5 ± 0.6 | 16.4 ± 0.9 | 17.3 ± 0.6 | 18.5 ± 0.5 |
| | Transformer w/ awake | 19.4 ± 0.4 | 16.7 ± 0.6 | 18.0 ± 0.6 | 19.9 ± 0.7 | 22.3 ± 0.6 |
| subject 4 (20.8 ± 0.4) | CNN w/o awake | 7.2 ± 0.2 | 20.6 ± 0.8 | 22.9 ± 0.9 | 25.2 ± 1.0 | 27.5 ± 0.9 |
| | CNN w/ awake | 8.1 ± 0.2 | 19.1 ± 0.7 | 21.7 ± 0.7 | 24.8 ± 0.6 | 26.7 ± 0.9 |
| | Transformer w/o awake | 16.6 ± 0.6 | 19.9 ± 0.7 | 22.1 ± 0.7 | 23.7 ± 0.9 | 24.4 ± 0.7 |
| | Transformer w/ awake | 18.0 ± 0.7 | 16.5 ± 0.5 | 18.8 ± 0.5 | 21.3 ± 0.8 | 22.2 ± 0.7 |
| subject 5 (9.5 ± 0.2) | CNN w/o awake | 6.5 ± 0.1 | 15.6 ± 0.9 | 18.2 ± 0.7 | 20.8 ± 0.7 | 22.4 ± 0.5 |
| | CNN w/ awake | 7.4 ± 0.1 | 13.0 ± 0.8 | 18.4 ± 0.7 | 22.4 ± 0.8 | 24.7 ± 0.9 |
| | Transformer w/o awake | 15.9 ± 0.7 | 15.6 ± 0.9 | 17.0 ± 0.7 | 18.2 ± 0.7 | 19.7 ± 0.6 |
| | Transformer w/ awake | 16.2 ± 0.6 | 13.5 ± 0.8 | 16.0 ± 0.7 | 17.2 ± 0.6 | 20.9 ± 0.7 |
| subject 6 (14.9 ± 0.7) | CNN w/o awake | 7.7 ± 0.2 | 10.5 ± 0.7 | 12.0 ± 0.7 | 14.9 ± 0.5 | 16.4 ± 0.9 |
| | CNN w/ awake | 7.3 ± 0.2 | 11.3 ± 0.9 | 15.0 ± 0.7 | 16.6 ± 0.6 | 19.7 ± 0.7 |
| | Transformer w/o awake | 13.6 ± 0.7 | 12.8 ± 0.9 | 14.3 ± 0.7 | 15.2 ± 0.6 | 17.3 ± 0.6 |
| | Transformer w/ awake | 14.7 ± 0.5 | 14.0 ± 0.7 | 16.0 ± 0.8 | 18.1 ± 0.6 | 21.4 ± 0.9 |
| subject 7 (15.7 ± 1.1) | CNN w/o awake | 7.3 ± 0.1 | 15.1 ± 0.4 | 18.6 ± 0.7 | 20.9 ± 0.9 | 22.1 ± 0.7 |
| | CNN w/ awake | 8.5 ± 0.2 | 20.9 ± 0.5 | 22.1 ± 0.8 | 23.4 ± 0.7 | 24.4 ± 0.7 |
| | Transformer w/o awake | 13.5 ± 0.6 | 16.3 ± 0.7 | 17.4 ± 0.6 | 18.6 ± 0.6 | 20.9 ± 0.8 |
| | Transformer w/ awake | 14.7 ± 0.7 | 14.5 ± 0.7 | 15.5 ± 0.5 | 17.2 ± 0.8 | 19.2 ± 0.7 |
| subject 8 (17.9 ± 0.8) | CNN w/o awake | 6.7 ± 0.1 | 17.3 ± 0.7 | 19.1 ± 0.9 | 20.9 ± 0.7 | 23.6 ± 0.7 |
| | CNN w/ awake | 6.5 ± 0.1 | 17.3 ± 0.8 | 23.3 ± 0.9 | 26.4 ± 1.1 | 28.2 ± 1.0 |
| | Transformer w/o awake | 15.7 ± 0.7 | 13.6 ± 0.9 | 15.5 ± 0.6 | 16.4 ± 0.5 | 20.0 ± 0.7 |
| | Transformer w/ awake | 16.1 ± 0.6 | 14.6 ± 0.8 | 17.2 ± 0.6 | 16.7 ± 0.6 | 17.6 ± 0.7 |
| subject 9 (11.5 ± 0.5) | CNN w/o awake | 6.6 ± 0.1 | 10.7 ± 0.7 | 11.5 ± 0.7 | 13.1 ± 0.6 | 13.9 ± 0.5 |
| | CNN w/ awake | 6.8 ± 0.1 | 14.8 ± 0.8 | 17.5 ± 0.6 | 19.2 ± 0.7 | 20.5 ± 0.7 |
| | Transformer w/o awake | 14.1 ± 0.2 | 15.6 ± 0.5 | 12.3 ± 0.8 | 11.5 ± 0.7 | 13.9 ± 0.7 |
| | Transformer w/ awake | 16.8 ± 0.3 | 13.5 ± 1.0 | 16.8 ± 0.4 | 18.9 ± 0.5 | 20.2 ± 0.7 |
| subject 10 (15.7 ± 1.1) | CNN w/o awake | 6.6 ± 0.1 | 15.0 ± 0.9 | 17.7 ± 0.7 | 19.6 ± 0.6 | 23.5 ± 1.2 |
| | CNN w/ awake | 6.5 ± 0.1 | 17.7 ± 0.9 | 20.4 ± 0.7 | 23.7 ± 0.4 | 25.5 ± 0.8 |
| | Transformer w/o awake | 15.2 ± 0.3 | 15.0 ± 0.6 | 16.3 ± 0.7 | 17.1 ± 0.7 | 17.7 ± 0.5 |
| | Transformer w/ awake | 15.3 ± 0.2 | 14.8 ± 1.1 | 15.6 ± 0.8 | 16.5 ± 1.1 | 18.4 ± 0.9 |
| subject 11 (18.2 ± 0.5) | CNN w/o awake | 6.1 ± 0.1 | 16.0 ± 1.1 | 20.2 ± 0.7 | 24.5 ± 0.7 | 26.7 ± 0.5 |
| | CNN w/ awake | 5.5 ± 0.1 | 21.3 ± 0.7 | 25.8 ± 0.9 | 27.3 ± 0.5 | 30.9 ± 1.4 |
| | Transformer w/o awake | 14.4 ± 0.4 | 16.0 ± 0.7 | 20.2 ± 0.8 | 21.3 ± 0.7 | 23.4 ± 0.9 |
| | Transformer w/ awake | 14.7 ± 0.2 | 14.3 ± 0.5 | 16.4 ± 0.5 | 18.1 ± 0.9 | 22.8 ± 0.7 |
| subject 12 (16.7 ± 0.8) | CNN w/o awake | 7.2 ± 0.1 | 11.9 ± 0.9 | 13.7 ± 0.7 | 18.5 ± 0.4 | 19.6 ± 0.7 |
| | CNN w/ awake | 5.4 ± 0.1 | 18.8 ± 0.6 | 21.6 ± 1.2 | 24.0 ± 0.9 | 25.0 ± 0.6 |
| | Transformer w/o awake | 15.8 ± 0.3 | 14.0 ± 0.7 | 16.1 ± 0.5 | 16.7 ± 1.0 | 17.3 ± 0.8 |
| | Transformer w/ awake | 16.0 ± 0.2 | 15.3 ± 0.8 | 17.0 ± 1.1 | 17.5 ± 0.9 | 19.7 ± 0.8 |

