# OpenReview forum: "Universal Sleep Decoder: Aligning awake and sleep neural representation across subjects"
_ICLR.cc/2024/Conference — Submitted to ICLR 2024_

### Official Review · Reviewer_i9wd · 2023-10-31

**Soundness:** 2 fair
**Presentation:** 2 fair
**Contribution:** 3 good
**Rating:** 6
**Confidence:** 4

**Summary:**

The authors present a dataset and a deep learning pipeline to decode evoked semantic categories during sleep. They collect a dataset of 64-channel EEG recorded while participants (n=52) were exposed to image and sound stimuli from 15 semantic categories while awake. Following this, participants were re-exposed to a subset of the previous sound stimuli while they were in N2/3 sleep. Deep learning pipelines based on CNNs or Transformers and combining a classification objective and a domain (i.e. awake-image, awake-sound, sleep-sound) adaptation objective were then trained to predict the category of a presented stimulus from the corresponding EEG. Different training and evaluation settings are investigated. Results suggest semantic categories can be decoded significantly above chance performance even during NREM sleep.

**Strengths:**

Originality: The proposed dataset, research question and decoding approach (combining classification and contrastive objectives for domain adaptation) appear to be novel.

Quality: The paper is of overall good quality and presents a complete picture of the research question, data and deep learning pipeline.

Clarity: The paper is overall clear, with the different components of the study and results exposed and mostly clearly described. See Weaknesses for proposed clarifications.

Significance: The study appears like an important step towards the understanding and improvement of semantic decoding during sleep. Along with the dataset (when released) this has the potential to effectively become a baseline framework for studying semantic decoding during sleep.

**Weaknesses:**

A core claim of the paper is that the experimental paradigm allows probing memory reactivation during sleep. However, I am not convinced the presented analyses actually allow studying memory reactivation. Rather, the trained neural encoders likely picked up on evoked activity related to the audio stimuli presented during sleep.
 First, the EEG recordings were epoched from -0.2 to 0.8 s around the stimulus onset (Section 4.1). The paper does not describe the distribution of audio stimuli duration, but it is likely that the audio clips lasted a few hundreds of milliseconds. In that case, the EEG windows likely contained evoked responses to these auditory stimuli rather than an associated memory. To assess that, an analysis of the evoked response that also takes into account the spatial dimension would be important (see Q1). On a related note, details of how auditory cues impacted sleep would be important to provide (Q2).
Second, I believe the data that was collected during sleep could be used to clarify this point. What kind of decoding performance can be achieved when only looking at auditory cues of mismatched pairs? If semantic category classification performance remains high for cues for which the evoked response should be different from the “memory-evoked” response (i.e. mismatched pairs), this could support the authors’ claim (see Q3).
Finally, to further support the claim that the paradigm tests memory reactivation, an analysis of the behavioral responses during the post-sleep session for presented vs. non-presented stimuli could be carried out. A significant increase in performance for stimuli presented during sleep could support the effect of the TMR-like protocol.
Overall, I believe these questions should be answered for the memory-related claims to be kept in the manuscript.

The description of the models in the Appendix is a bit confusing (see Q4). Summarizing the entire architecture (i.e. including more than just the Conv layers in the table) would be helpful. Also, a single description of the “Subject block” might be clearer (instead of having two separate tables that appear to contain the same information).

**Questions:**

1. What do the evoked responses look like? It would be important to provide descriptive analyses of the time-locked response to images, auditory cues and auditory cues during N2/3 to confirm the validity of the collected data. Importantly, do time-locked responses during sleep follow a different temporal pattern that maybe spans a longer window (as memory reactivation might happen after the stimulus presentation)? Moreover, considering the spatial dimension of the evoked response (i.e. how it is distributed across the different EEG channels, e.g. with topomaps) might help confirm the responses collected during sleep are actually closer to (awake) auditory or visual responses.

2. Is there a chance the audio cues during N2/3 woke up the participants? Showing examples and/or a summary of sleep staging (e.g. hypnograms showing how N2/3 stages were not interrupted by the cues) would be useful.

3. How does decoding performance during sleep differ for auditory cues coming from matched vs. mismatched pairs? My understanding from Section A2 is that the sleep auditory cues were randomly selected from the whole audio set, meaning there should be examples from both matched and mismatched pairs available. A supplementary figure like Figure 3 could then be used to report the results for both categories. If performance remains as high for auditory cues of mismatched pairs, then the “memory-replay” hypothesis might be validated.

4. In Section 2.3: “Since there are fewer publicly available EEG recordings during sleep compared to those during wakefulness, applying unsupervised pretraining methods for sleep decoding is not feasible.” I believe that is not true, as there are a lot of openly available sleep datasets (SHHS, MASS, SleepEDF, Physionet Challenge 2018, etc.). My understanding is the limiting factor might be the spatial coverage for those datasets though, which often include a few channels only whereas the presented dataset contains 64 channels.

5. What is the impact of the hyperparameter $\lambda$ in Equation 2, and how was the value of 0.5 selected in Section 4.2.3?

6. The performance of the Lasso GLM is about the same as the neural decoders in Figure 3c. How does the Lasso GLM fare in the Awake+Sleep → Sleep setting (Figure 3d)?

7. In Section 4.2.1: “We take the test accuracy according to the maximum validation accuracy as its performance.” I am not sure I understand what this means.

8. Use of the word “migration” (e.g. Section 4.2.2): maybe “transfer” would be clearer and more connected with the literature?

---

> ### Author Response · Authors · 2023-11-18
> **Response to Reviewer i9wd (1/3)**
>
> Thank you for the help review. Below, we give a point-by-point response:
>
> ## Weakness 1:
> A core claim of the paper is that the experimental paradigm allows probing memory reactivation during sleep. However, I am not convinced the presented analyses actually allow studying memory reactivation. Rather, the trained neural encoders likely picked up on evoked activity related to the audio stimuli presented during sleep. First, the EEG recordings were epoched from -0.2 to 0.8 s around the stimulus onset (Section 4.1). The paper does not describe the distribution of audio stimuli duration, but it is likely that the audio clips lasted a few hundreds of milliseconds. In that case, the EEG windows likely contained evoked responses to these auditory stimuli rather than an associated memory. To assess that, an analysis of the evoked response that also takes into account the spatial dimension would be important (see Q1).
> ## Response:
> Sorry for the confuse. We supplemented the extra results of Lasso GLM on sleep->awake (including sleep->image and sleep->audio) experiments. We can observe that the accuracy of Lasso GLM, on either awake->sleep experiments (see Fig.3(c)) or sleep->awake experiments (see Fig.8 in Appendix), is significantly higher than random levels.
>
> Besides, all audio stimulus during sleep last for 0.5s. The audio stimulus used during sleep are the same with those used during wakefulness. We have revised the experimental design related to the sleep portion in Appendix A, emphasizing that the duration of the all audios played during sleep is 0.5 second.
>
> ## Question 1:
> What do the evoked responses look like? It would be important to provide descriptive analyses of the time-locked response to images, auditory cues and auditory cues during N2/3 to confirm the validity of the collected data. Importantly, do time-locked responses during sleep follow a different temporal pattern that maybe spans a longer window (as memory reactivation might happen after the stimulus presentation)? Moreover, considering the spatial dimension of the evoked response (i.e. how it is distributed across the different EEG channels, e.g. with topomaps) might help confirm the responses collected during sleep are actually closer to (awake) auditory or visual responses.
> ## Response:
> Thank you for bringing this up, and it's indeed a crucial question. We separately computed the averaged event-related potentials (ERPs) induced by image and audio stimuli during wakefulness and TMR-related audio stimuli during sleep, along with their topographical maps. As you noted, the temporal patterns of stimulus ERPs during sleep are indeed markedly different, spanning longer windows compared to wakefulness. These results have been added to the Appendix Fig.6.
>
> Furthermore, in order to assess the spatial dimensions of TMR-related ERPs and their similarity to auditory or visual ERPs during wakefulness, we initially identified the time point with the maximum variability in the averaged ERPs for image and audio across all participants. Each participants’ spatial distributions of ERPs at these two time points for image and audio were considered as the templates. Subsequently, the spatial correlation between the TMR-related ERPs at each time point and the two templates for each participant was computed. Finally, paired t-tests were conducted on the two sets of correlation coefficients at each time point, and significant differences were annotated. The results indicate that the spatial features (topomaps) of TMR-related ERPs are significantly closer to the wakeful period image stimulus template at 0.56-1.00 seconds and significantly closer to the wakeful period audio stimulus template at 1.29-1.42 seconds. The result have been added to the Appendix Fig.7.
>
> ## Question 2:
> Is there a chance the audio cues during N2/3 woke up the participants? Showing examples and/or a summary of sleep staging (e.g. hypnograms showing how N2/3 stages were not interrupted by the cues) would be useful.
> ## Response:
> Thank you for the suggestion. Actually, we confirmed the appropriate sound volume during the pre-experimental period, ensuring that participants were almost never awakened by the TMR sound stimuli during N2/3 sleep stages. And we have compiled the entire night's sleep stages, spectrograms, and TMR-related audio cue time points for three participants as examples, and have added them to the Appendix Fig.5.

---

> ### Author Response · Authors · 2023-11-18
> **Response to Reviewer i9wd (2/3)**
>
> ## Weakness 2 & Question 3:
> Second, I believe the data that was collected during sleep could be used to clarify this point. What kind of decoding performance can be achieved when only looking at auditory cues of mismatched pairs? If semantic category classification performance remains high for cues for which the evoked response should be different from the “memory-evoked” response (i.e. mismatched pairs), this could support the authors’ claim (see Q3).
>
> Finally, to further support the claim that the paradigm tests memory reactivation, an analysis of the behavioral responses during the post-sleep session for presented vs. non-presented stimuli could be carried out. A significant increase in performance for stimuli presented during sleep could support the effect of the TMR-like protocol. Overall, I believe these questions should be answered for the memory-related claims to be kept in the manuscript.
>
> How does decoding performance during sleep differ for auditory cues coming from matched vs. mismatched pairs? My understanding from Section A2 is that the sleep auditory cues were randomly selected from the whole audio set, meaning there should be examples from both matched and mismatched pairs available. A supplementary figure like Figure 3 could then be used to report the results for both categories. If performance remains as high for auditory cues of mismatched pairs, then the “memory-replay” hypothesis might be validated.
> ## Response:
> We apologize for not clearly describing the experimental design. In fact, there were no mismatched stimulus pairs in our experiment; we used 15 images and 15 audio as the stimulus, and all stimulus were paired during wakefulness. Additionally, we did not use the classical TMR experimental design. Instead, all the image-audio stimulus pairs used in the experiment were presented with equal probability during sleep. This is why we refer to stimuli during N2/N3 sleep as TMR-related stimuli.
>
> ## Question 4:
> In Section 2.3: “Since there are fewer publicly available EEG recordings during sleep compared to those during wakefulness, applying unsupervised pretraining methods for sleep decoding is not feasible.” I believe that is not true, as there are a lot of openly available sleep datasets (SHHS, MASS, SleepEDF, Physionet Challenge 2018, etc.). My understanding is the limiting factor might be the spatial coverage for those datasets though, which often include a few channels only whereas the presented dataset contains 64 channels.
> ## Response:
> Yes, you are right. Integrating different EEG datasets with different sampling channel configuration (e.g., 16, 32, 64 channels) is still an ongoing research area. There are a lot of studies [1] trying to solve this challenge using awake datasets. And we changed this sentence to “Since there are fewer publicly available EEG recordings during sleep compared to those during wakefulness, applying unsupervised pretraining methods for sleep decoding is still under exploration.”
>
> [1] Yi K, Wang Y, Ren K, et al. Learning Topology-Agnostic EEG Representations with Geometry-Aware Modeling[C]//Thirty-seventh Conference on Neural Information Processing Systems. 2023.
>
> ## Weakness 3:
> The description of the models in the Appendix is a bit confusing (see Q4). Summarizing the entire architecture (i.e. including more than just the Conv layers in the table) would be helpful. Also, a single description of the “Subject block” might be clearer (instead of having two separate tables that appear to contain the same information).
> ## Response:
> Sorry for the confuse. We have summaried the entire model into one table, see Table.1-4 in Appendix for more details. Updated pdf is uploaded. And we provide more details about “Subject Block” in the Apendix C.2.1.

---

> ### Author Response · Authors · 2023-11-18
> **Response to Reviewer i9wd (3/3)**
>
> ## Question 5:
> What is the impact of the hyperparameter $\lambda$ in Equation 2, and how was the value of 0.5 selected in Section 4.2.3?
> ## Response:
> We provide an additional figure about the effect of $\lambda$ over the sleep decoding performance in Appendix Fig.12, which is done during waiting for the reviews. We can see that $\lambda=0.5$ performs the best. A setting of $\lambda=0$ leads to the loss of the ability to align awake and sleep data in the representation space, while $\lambda=10$ results in a relatively low proportion of classification loss, making it difficult to learn discriminative features. Of course, $\lambda=0.5$ may not be the optimal setting, as we have not conducted an exhaustive search across the hyperparameter space.
>
> ## Question 6:
> The performance of the Lasso GLM is about the same as the neural decoders in Figure 3c. How does the Lasso GLM fare in the Awake+Sleep → Sleep setting (Figure 3d)?
> ## Response:
> Sorry for missing this figure. We provided an additional figure about the decoding performance of Lasso GLM in the Awake+Sleep->Sleep setting in Appendix Fig.9, which is done before submission. We can see that adding additional awake data will slightly decrease the performance (-0.1%), due to the large gap of neural patterns between awake data and sleep data. This result correspond the setting of $\lambda=0$ in CNN and Transformer model. However, as Lasso GLM directly maps the brain activity of one time point to the corresponding labels with only one layer, it’s hard to apply contrastive learning method, which is mainly developped for deep learning methods, to Lasso GLM.
>
> ## Question 7:
> In Section 4.2.1: “We take the test accuracy according to the maximum validation accuracy as its performance.” I am not sure I understand what this means.
> ## Response:
> Sorry for the confuse. In the individual training setting, we split the whole dataset of that subject into training (80%), validation (10%), test (10%) splits. Then, we train the model using the training split for 200 epochs. For each epoch, the train accuracy, validation accuracy, test accuracy is reported. We find the max validation accuracy from these 200 epochs. And **use the test accuracy at the same epoch as the performance, to avoid artificially high accuracy compared to direct selection of max test accuracy**, which is a common practice in deep learning. In addition, we **averaged the test accuracy over five different seeds to mitigate the impact of fluctuations**.
>
> ## Question 8:
> Use of the word “migration” (e.g. Section 4.2.2): maybe “transfer” would be clearer and more connected with the literature?
> ## Response:
> Thanks for your suggestion. The term “transfer” is indeed better. As awake data and sleep data can be viewed as different domains, the term 'transfer' aligns more closely with the definition associated with domain generalization in the field of computer science.

---

> > ### Comment · Reviewer_i9wd · 2023-11-21
> >
> > Thanks to the authors for their answers.
> >
> > Overall, I think the authors should clarify their experimental protocol and update the manuscript accordingly: it is still not clear whether matched/mismatched pairs were used, and how many examples were shown during data collection.
> >
> > If there are indeed no mismatched pairs, I currently don’t see clear evidence for the decoding of memory replay. As such, I would recommend the authors rephrase the claims on memory replay and focus on decoding of specific auditory instances during sleep.
> >
> > Please find some additional comments on specific questions below.
> >
> > [Q1] Thank you for providing the evoked potential figures. It is very interesting to see how different the response to auditory cues is during sleep - the large negative deflection at t=0.74 seems to be in the same frequency range as slow waves.
> >
> > [Q3] If I understand correctly then, Figure 1 and its caption, as well as the first paragraph of Section A.1 should be changed, or does inconsistent/consistent not mean the same as matched/mismatched here?
> >
> > > “In each image-audio pair, the picture and the sound may be inconsistent. In total, we had 210 inconsistent trials and 390 consistent trials.”
> >
> > Also, can you confirm there are only 15 different sounds and 15 different images that were used? Section A.1 indicates that there are 600 image-audio pairs so I assumed those would be 600 different images and 600 different audio clips.
> >
> > Overall, given pairs were actually matched (and that there were only 15 different audio clips), I don’t see clear evidence for the focus of the paper on memory replay, for instance as mentioned in the first sentence of the abstract: “Decoding memory content from brain activity during sleep has long been a goal in neuroscience.”
> >
> > Simply put, can a classifier trained on the awake EEG to predict one of 15 categories actually perform well on sleep data by just picking up on the evoked activity itself, rather than a “memory activated” response?
> >
> > My understanding is that a different data collection paradigm would be necessary to support the “memory replay” hypothesis, e.g. by (1) having mismatched pairs or (2) a control condition where a new instance/category is introduced during sleep. In (1), for instance, if the picture of a sheep is matched with the sound of a soccer ball, the decoder should be able to correctly predict “sheep” when the sound of a soccer ball is played back during sleep.
> >
> > Finally, I am concerned that if you used only 15 example image-audio pairs, the results are limited to instance-level classification, and do not support across category generalization.
> >
> > [W3] Thank you for the updated description of the architectures!

---

> > > ### Author Response · Authors · 2023-11-22
> > > **Response to Reviewer i9wd (4/4)**
> > >
> > > Thank you for the help review.
> > >
> > > > Overall, I think the authors should clarify their experimental protocol and update the manuscript accordingly: it is still not clear whether matched/mismatched pairs were used, and how many examples were shown during data collection.
> > > ## Response 1:
> > > Here's a rephrased explanation of the experimental design:
> > > Our experiment is a TMR-related study where, during the wakeful period, participants were presented with a set of 15 distinct images paired with 15 different sounds. This was done to ensure the establishment of stable associations between the 15 pairs of stimuli in the participants' memory. Specifically, participants engaged in a paired-associate memory task during wakefulness. They were presented with one randomly selected image from the set of 15 and one randomly selected sound from the set of 15. Participants were then required to answer  whether these two stimuli belong to the same pair. This process was repeated with the order of presentation switched between images and sounds. In this phase of the experiment, there were instances where the presented image and sound did not belong to the same stimulus pair. This is what we referred to as "inconsistent" stimulus pairs. However, the purpose of this was solely to confirm that participants had formed correct and stable stimulus pair memories. Next, we presented TMR-related stimuli during sleep. Specifically, after participants entered a stable N2/3 sleep stage, we randomly selected one of the 15 auditory stimuli to play every 4-6 seconds. More details can be found in Appendix A.
> > >
> > > > If there are indeed no mismatched pairs, I currently don’t see clear evidence for the decoding of memory replay. As such, I would recommend the authors rephrase the claims on memory replay and focus on decoding of specific auditory instances during sleep.
> > >
> > > > Simply put, can a classifier trained on the awake EEG to predict one of 15 categories actually perform well on sleep data by just picking up on the evoked activity itself, rather than a “memory activated” response?
> > >
> > > > Finally, I am concerned that if you used only 15 example image-audio pairs, the results are limited to instance-level classification, and do not support across category generalization.
> > > ## Response 2:
> > > Similar to what you mentioned, indeed, we were not decoding memory replay. In fact, we were decoding responses and memory content evoked by auditory stimuli during sleep. Differentiating between these two is indeed challenging. However, through the results of the Lasso-GLM, we have demonstrated that using only awake-period image stimuli data for training, decoding accuracy for audio stimuli during sleep is significantly higher than chance level. Conversely, training solely with sleep-period audio stimuli results in decoding accuracy for image stimuli during wakefulness that is significantly higher than chance level. These outcomes suggest that TMR-related sound stimuli during sleep do induce associated memory content to some extent.
> > > And indeed, category generalization is not supported in our current sleep decoding framework. Currently, decoding during sleep is limited to a close-set of stimuli. To achieve category generalization in sleep decoding, an open-set experiment that can be conducted during sleep would be necessary. However, as demonstrated in [1], zero-shot (or open-set) classification is indeed a more difficult task than supervised close-set classification. As presenting the subjects with 16000+ different pictures (each repeating 4 times), i.e. THINGS-EEG, the performance is still lower than directly training on left-out dataset (which contains 50 images, each repeating 80 times). Implementing such a design is inherently difficult, and we consider it a potential direction for future research goals.
> > >
> > > [1] Du C, Fu K, Li J, et al. Decoding visual neural representations by multimodal learning of brain-visual-linguistic features[J]. IEEE Transactions on Pattern Analysis and Machine Intelligence, 2023.

---

> > > > ### Comment · Reviewer_i9wd · 2023-11-22
> > > >
> > > > Thank you for the clarifications. Just to make sure, I don’t see changes to the manuscript, has it been updated? If I understand correctly, Figure 1, Section 4.1 and Appendix A still describe a different experimental paradigm.
> > > >
> > > > > However, through the results of the Lasso-GLM, we have demonstrated that using only awake-period image stimuli data for training, decoding accuracy for audio stimuli during sleep is significantly higher than chance level. Conversely, training solely with sleep-period audio stimuli results in decoding accuracy for image stimuli during wakefulness that is significantly higher than chance level. *These outcomes suggest that TMR-related sound stimuli during sleep do induce associated memory content to some extent.*
> > > >
> > > > I understand the authors’ point that the awake→sleep and sleep→awake results show that there is two-way transferability between awake image- and asleep audio-evoked responses. However, I don’t see evidence for these results to be related to *memory content*. An alternative hypothesis is that the decoders picked up on modality-independent, high-level semantic information (e.g. “sheep” or maybe just “animal”). This wouldn’t require any memory reactivation component and could explain the results of Figure 3(c).

---

> > > > > ### Author Response · Authors · 2023-11-22
> > > > > **Response to Reviewer i9wd (5/5)**
> > > > >
> > > > > We are deeply grateful to you for the thorough review and constructive comments, which have greatly assisted us in improving the quality and presentation of our manuscript.
> > > > >
> > > > > > Just to make sure, I don’t see changes to the manuscript, has it been updated? If I understand correctly, Figure 1, Section 4.1 and Appendix A still describe a different experimental paradigm.
> > > > > ## Response 1:
> > > > > We have unified these three descriptions of the experimental paradigm as much as possible, and have shown them in red in the updated document.
> > > > >
> > > > > > I understand the authors’ point that the awake→sleep and sleep→awake results show that there is two-way transferability between awake image- and asleep audio-evoked responses. However, I don’t see evidence for these results to be related to memory content. An alternative hypothesis is that the decoders picked up on modality-independent, high-level semantic information (e.g. “sheep” or maybe just “animal”). This wouldn’t require any memory reactivation component and could explain the results of Figure 3(c).
> > > > > ## Response 2:
> > > > > We agree with the reviewer that this study cannot distinguish the specific memory content from its corresponding high-level semantics. In our understanding, they are the same thing. We are not intending to differentiate them. To make it clear, we have made changes to the paper in the Conclusion & limitations section, also shown in red.

---

> > > > > > ### Comment · Reviewer_i9wd · 2023-11-22
> > > > > >
> > > > > > Thanks again for the additional answers.
> > > > > >
> > > > > > > We agree with the reviewer that this study cannot distinguish the specific memory content from its corresponding high-level semantics. In our understanding, they are the same thing. We are not intending to differentiate them.
> > > > > >
> > > > > > My understanding is that they are not the same, though I might be wrong. I would be interested in seeing references that support the equivalence between activity evoked by memory content and by high-level semantics if there are any you can provide.
> > > > > >
> > > > > > Based on the answers given during the discussion period (1) I believe the scope of the dataset and results is more limited than initially thought and (2) the focus of the manuscript on memory-related processes is still unclear. Given the additional technical clarifications and supplementary analyses though I conserve my initial score.

---

> > > > > > > ### Author Response · Authors · 2023-11-23
> > > > > > > **Response to Reviewer i9wd (6/6)**
> > > > > > >
> > > > > > > > My understanding is that they are not the same, though I might be wrong. I would be interested in seeing references that support the equivalence between activity evoked by memory content and by high-level semantics if there are any you can provide.
> > > > > > > ## Response:
> > > > > > > We have mentioned that some TMR studies in Related Work section (i.e., section 2.1). And their results have shown that TMR stimulus during sleep can indeed evoke high-level semantics memory.
> > > > > > >
> > > > > > > > However, several studies in recent years indicate that sleepers can process external stimuli at different levels of cognitive representation, encompassing semantic and decision-making stages, rather than merely at the level of low-level sensory processing (Strauss et al., 2015; Issa & Wang, 2011; Kouider et al., 2014). Furthermore, learning-related sensory cues presented during sleep positively impact subsequent recall of cue-related material upon awakening (Rasch et al., 2007; Hu et al., 2020), which is commonly referred to as Target Memory Reactivation (TMR).

---

### Official Review · Reviewer_mxAy · 2023-10-31

**Soundness:** 2 fair
**Presentation:** 4 excellent
**Contribution:** 3 good
**Rating:** 6
**Confidence:** 4

**Summary:**

The paper reports on a novel EEG dataset with both sleeping and awake participants designed for memory reactivation decoding during sleep. In addition, it provides a comprehensive set of competitive baselines and ablations on this data demonstrating within-participant and cross-participant generalization.

**Strengths:**

The paper is clearly written, and the contributions are clearly detailed and (mostly) well supported. Furthermore the paper promises to release a novel dataset (in the supplementary material) and provides clean and reasonably well-annotated code for its contributions. The set of experiments is comprehensive.

**Weaknesses:**

My primary concern in reading the paper is that a core contribution regarding the relative performance of the various models is not supported as well as it could be. In particular, I think we should expect something like performance on awake+sleep+contrastive > awake+sleep > [awake or sleep in whatever order]. This data is all available in plots (Fig 3) but from just eyeballing those plots it's hard to tell whether the sleep->sleep CNN is better or worse than the awake+sleep->sleep CNN, for example. These should ideally be on the same plot. Similarly sections 4.2.2 and 4.2.3 report some statistics but no comparison to support the core claim above, and there's no results table for these experiments either, unless I have missed something.

Separately, I have some notation concerns:
* Is it really that $y \in \mathbb{R}^K$, i.e. each label is a vector of real numbers the length of the number of classes? I would think it's $y \in \\{1 \ldots K\\}$ or similar.
* The $\mathcal{X}$s aren't explicitly defined.
* If $\mathcal{P}(i) = \\{k|k\in \mathcal{A}(i), y_k = y_i\\}$ and $\mathcal{A}(i)$ is a set of instances $\\{x_i, y_i\\}$ then $k$ is such an instance and $y_k$ seems overloaded or poorly defined.

These are just the ones I immediately caught -- another careful proofread of the math might be useful.

Finally, some more unordered comments:
* I think describing a dataset as "open set" is a bit odd (section 2.2) -- a dataset has a fixed number of classes, i.e. it is "closed set". In my understanding "open set" is a notion w.r.t models / tasks rather than datasets (i.e. ability to classify unseen classes, often by composition of seen classes, the use of a language model, or something else).
* Kostas et al. 2021 (doi:10.3389/fnhum.2021.653659) is likely worth mentioning in discussion of larger-scale SSL pretraining for EEG.
* I found Figure 2 more confusing than illustrative -- the caption is doing a lot of explanation and I'm not sure how much the figure adds. For example, the arrows and colors are not used consistently -- it's not obvious what the arrow colors mean, and the arrows seem to indicate data flow in the top part of the figure and an ordering of experiments in the bottom part.

**Questions:**

* The two plots in Figure 4 have the same y axis (which should facilitate comparison) but if I understand things correctly, the x axes indicate percentages of different amounts of data (even though each batch is balanced). Is that right? If so, maybe this plot's x axis should be the number of instances / hours etc to facilitate direct comparison.
* The paper takes care to describe its paradigm as "TMR related" instead of "TMR evoked" because the paradigm is different from TMR -- how is it different?

**Details Of Ethics Concerns:**

The paper does not discuss informed consent and/or external oversight/review/approval (IRB or similar) which I imagine should be required for this work. Data release should have also been reviewed separately and anonymization protocols defined and approved. I imagine that omitting this information is just an oversight and things were done appropriately, but wanted to flag for verification.

---

> ### Author Response · Authors · 2023-11-18
> **Response to Reviewer mxAy (1/2)**
>
> Thank you for the help review. Below, we give a point-by-point response:
>
> ## Weakness 1:
> My primary concern in reading the paper is that a core contribution regarding the relative performance of the various models is not supported as well as it could be. In particular, I think we should expect something like performance on awake+sleep+contrastive > awake+sleep > [awake or sleep in whatever order]. This data is all available in plots (Fig 3) but from just eyeballing those plots it's hard to tell whether the sleep->sleep CNN is better or worse than the awake+sleep->sleep CNN, for example. These should ideally be on the same plot. Similarly sections 4.2.2 and 4.2.3 report some statistics but no comparison to support the core claim above, and there's no results table for these experiments either, unless I have missed something.
> ## Response:
> Yes, the ideal result would be awake+sleep+contrastive > awake+sleep > sleep, as we mainly introduce awake data to improve sleep decoding. However, there is a large gap between neural patterns of awake and sleep data, (as demonstrated in Appendix Fig.6).
>
> Directly using a balanced batch (comprising half awake and half sleep data), does not exhibit a significant improvement on the sleep decoding performance. In some cases, there was even a slight decrease in model performance (~0.1%), primarily due to the issues mentioned before. We conducted a control experiment with CNN model, shuffling labels for awake data and inputting them alongside sleep data into the model. This significantly reduced the model's performance, see Appendix Fig.11 for more details. Besides, detailed performance (mean/std) of different models are shown in Appendix Table5-6.
>
> ## Weakness 2-4:
> So sorry for the mistakes. We have conducted a careful proofread of math, Updated pdf is uploaded.
> ### Weakness 2:
> Is it really that $y\in\mathbb{R}^{K}$, i.e. each label is a vector of real numbers the length of the number of classes? I would think it's $y\in\\{1,...,K\\}$ or similar.
> ### Response:
> Sorry for the mistake. Yes, it’s $y\in\\{1,...,K\\}$. $y\in\mathbb{R}^{K}$ is just a one-hot vector corresponding to $y\in\\{1,...,K\\}$.
> ### Weakness 3:
> The $\mathcal{X}$s aren’t explicitly defined.
> ### Response:
> Sorry for the mistake. $\mathcal{X}^{img}$ means all image-evoked EEG recordings of one subject, i.e., $\mathcal{X}^{img}=\\{x_{n}^{img}\\}_{n=1}^{N^{img}}$. $\mathcal{X}^{aud}$ and $\mathcal{X}^{tmr}$ are similarly defined.
> ### Weakness 4:
> If $\mathcal{P}(i)=\\{k|k\in\mathcal{A}(i),y_{k}=y_{i}\\}$ and $\mathcal{A}(x)$ is a set of instances $\\{x_{i},y_{i}\\}$ then $k$ is such an instance and $y_{k}$ seems overloaded or poorly defined.
> ### Response:
> Sorry for the mistake. We use $\\{1,...,|\mathcal{B}|\\}$ to indicate the indices of $\mathcal{B}$. For example:
>
> $\mathcal{A}(i) = \\{1,...,|\mathcal{B}|\\} \setminus \{i\}$
>
> Due to display issues on the web page, you can find the updated formula of the whole contrastive loss in the updated pdf.
>
> ## Weakness 5:
> I think describing a dataset as "open set" is a bit odd (section 2.2) -- a dataset has a fixed number of classes, i.e. it is "closed set". In my understanding "open set" is a notion w.r.t models / tasks rather than datasets (i.e. ability to classify unseen classes, often by composition of seen classes, the use of a language model, or something else).
> ## Response:
> Sorry for the mistake. We have made modificatiosn to the relevant terminologies, e.g., dataset with a considerable number of categories, dataset with few categories. We apologize for not being able to provide a more precise definition; we can only delineate the categories of the dataset in a relatively vague manner.

---

> ### Author Response · Authors · 2023-11-18
> **Response to Reviewer mxAy (2/2)**
>
> ## Weakness 6:
> Kostas et al. 2021 (doi:10.3389/fnhum.2021.653659) is likely worth mentioning in discussion of larger-scale SSL pretraining for EEG.
> ## Response:
> We sincerely apologize for the omission of the crucial reference to the BENDR paper. In BENDR, the primary focus is multivariate mask modeling. In DreamDiffusion, a similar approach is employed (with a slight modification when embedding subseries-level patches).
>
> ## Weakness 7:
> I found Figure 2 more confusing than illustrative -- the caption is doing a lot of explanation and I'm not sure how much the figure adds. For example, the arrows and colors are not used consistently -- it's not obvious what the arrow colors mean, and the arrows seem to indicate data flow in the top part of the figure and an ordering of experiments in the bottom part.
> ## Response:
> Sorry for the confuse. We have removed those confusing arrows. Updated figure (Fig.2 in main text) is uploaded.
>
> ## Question 1:
> The two plots in Figure 4 have the same y axis (which should facilitate comparison) but if I understand things correctly, the x axes indicate percentages of different amounts of data (even though each batch is balanced). Is that right? If so, maybe this plot's x axis should be the number of instances / hours etc to facilitate direct comparison.
> ## Response:
> Sorry for the confuse. As every sleep data item is paired with one awake-image data item and one awake-audio data item (randomly selected from the awake dataset from the same subject), the amounts of finetune data used in awake&sleep pretrained model is three times the amounts of finetune data used in sleep pretrained model. But the amounts of sleep data used in finetuning stage are equal. We have modified the figure caption to emphasize the distinction on the x-axis between Fig.4(a) and Fig.4(b). However, the label on the x-axis should remain in percentage form, as the data sizes vary for each finetuning individual, and using percentages provides a standardized representation. The caption of Fig.4 is updated.
>
> ## Question 2:
> The paper takes care to describe its paradigm as "TMR related" instead of "TMR evoked" because the paradigm is different from TMR -- how is it different?
> ## Response:
> We apologize for not clearly describing the experimental design. In the classical TMR experimental design, only a subset of the image-audio stimulus pairs are presented during sleep to induce memory reactivation. The remaining pairs of image-audio stimulus are not presented during sleep, and hence, are referred to as "targeted" memory reactivation. Instead, we did not use the classical TMR experimental design, all the image-audio stimulus pairs in our experiment were presented with equal probability during sleep. This is why we describe our paradigm as "TMR related".
>
> ## Details Of Ethics Concerns:
> The paper does not discuss informed consent and/or external oversight/review/approval (IRB or similar) which I imagine should be required for this work. Data release should have also been reviewed separately and anonymization protocols defined and approved. I imagine that omitting this information is just an oversight and things were done appropriately, but wanted to flag for verification.
> ## Response:
> Thanks for your reminder. IRB information is provided in **Ethics statement**.

---

> > ### Comment · Reviewer_mxAy · 2023-11-22
> > **Thank you for your detailed responses**
> >
> > I thank the authors for the detailed responses, I think the clarity of the paper is improved. However, from examining appendix tables 5 and 6, I see that there is no gap between models trained on awake+sleep data vs models trained on sleep data only, and I'm not sure I find the label-shuffling ablation convincing (why should we not expect label-shuffled training to worsen performance?). As things stand it seems like contrastive pretraining is what is providing the advantage (rather than the awake data). I imagine this could be verified by using a contrastive-only pretraining model, and it places doubt on the claim that awake-sleep alignment is what is doing the work (vs just the availability of some sort of data for contrastive pretraining). Based on this I will keep my rating. Regardless of whether the final decision is acceptance or a rejection, I recommend that the authors reframe the next draft to emphasize the benefits of contrastive pretraining, vs sleep-awake "alignment".

---

> > > ### Author Response · Authors · 2023-11-23
> > > **Response to Reviewer mxAy (3/3)**
> > >
> > > Thank you for the helpful review.
> > >
> > > ## Response:
> > > To make it clear, awake-sleep alignment is exactly what contrastive learning is trying to do:
> > >  - **We use supervised contrastive learning, which is trying to minimize the distance between the latents of awake and sleep data.**
> > >  - In this study, we never use the presented pictures and sounds to get the corresponding embeddings, thus doing contrastive learning. **Only EEG signals, nothing else.**
> > >
> > > “Label shuffling” is trying to demonstrate that directly incorporating sleep data with unrelated data items decreases the performance. When you train a classifier, any additional unrelated data items leads to the drop of performance. Compared to “label shuffling” one, directly incorporating sleep data with awake data items doesn’t show great drop, which exactly means that they share the common semantics. What we have to do is to use some methods to reduce the gap between awake data and sleep data. That’s why we use contrastive learning to reduce the domain gap between these two domains. Therefore, **awake-sleep alignment is exactly what contrastive learning is trying to do**.

---

### Official Review · Reviewer_BG5d · 2023-11-06

**Soundness:** 2 fair
**Presentation:** 2 fair
**Contribution:** 2 fair
**Rating:** 3
**Confidence:** 5

**Summary:**

This paper introduces an approach for decoding memory content from brain activity during sleep. To be more specific, the authors show an experimental setup to extract memory reactivation during NREM sleep, along with the ground truth timing and content during the neural replay episode. Using the dataset from 52 subjects, they train a model capable of generalizing across subjects in a zero-shot manner.

**Strengths:**

The data collected for the paper could be useful for researchers in biosignals/sleep community.

**Weaknesses:**

- The data is not released. Without the data it's difficult to verify the claims, since one of the main claims of the paper seems to be the unique data collected. Since this appears to be a dataset paper, it is essential that the data is released and verified before this can be accepted.
- There is no comparison with other works. Without the dataset, this paper does not contribute much else.
- With such low accuracy as shown in Figure 3, the efficacy of the method is put into doubt.
- What are the asterisks in Figure 3?
- Even if the data is released, other venues (more focused on health/physiological signals) would be suitable for this paper.

**Questions:**

Please address the weaknesses.

**Details Of Ethics Concerns:**

- Human subjects research: Data collection process does not have any real details. IRB information is not provided.
- Annotation process is not provided. If renumeration or anything else was provided as part of participants/annotators.

---

> ### Author Response · Authors · 2023-11-18
> **Response to Reviewer BG5d (1/2)**
>
> Thank you for your suggestions. Below, we give a point-by-point response:
>
> ## Weakness 1:
> The data is not released. Without the data it's difficult to verify the claims, since one of the main claims of the paper seems to be the unique data collected. Since this appears to be a dataset paper, it is essential that the data is released and verified before this can be accepted.
> ## Response:
> The dataset is still under collection, targeting about 100+ available subjects. Thus, we can provide a large enough benchmark dataset for sleep decoding. And the dataset will be released by the end of this year (2023). **To verify the claims, we provide a demo dataset (preprocessed and anonymized) containing 3 randomly selected subjects. You can download the demo-dataset from [here](https://drive.google.com/drive/folders/1013NKON4U9D8E60OdLbImwRfvoGlPlaE?usp=sharing).** You can directly run the code, after creating the conda environment, see README.md for more details.
>
> ## Weakness 2:
> There is no comparison with other works. Without the dataset, this paper does not contribute much else.
> ## Response:
> As there is no previous study that has collected such a large sleep decoding dataset, previous studies mainly uses conventional machine learning algorithms, e.g., lasso GLM, instead of deep learning models to build a sleep decoder. Not to mention the alignment between awake data and sleep data. To our knowledge, we are the first to collect such a large and high-density dataset, and apply advanced deep learning methods (e.g., supervised pretraining, contrastive learning) to the semantic sleep decoding problem.
>
> ## Weakness 3:
> With such low accuracy as shown in Figure 3, the efficacy of the method is put into doubt.
> ## Response:
> It’s worth noting that we are doing sleep decoding with EEG recordings, which is much harder than decoding images from EEG recordings, as demonstrated in Fig.3(a). In the current dataset, we have tried to collect and train sleep decoders of 15 distinct auditory cue- visual image pairs. This is significantly much larger than any TMR studies we know of, in the field of neuroscience [1-2]. This will lay foundations for future sleep decoding studies.
>
> When decoding images from EEG recordings, we achieve about 49.5% on 15-way classification task (with random level 6.7%). In EEGNet [3], with P300 dataset, it achieves about 90% on 2-way classification task (with random level 50%).
>
> Moreover, we further improve the sleep decoding performance by finetuning pretrained model (using other subjects) on that subject to 25.9% on average (over 12 subjects) as demonstrated in Fig.4(b).
>
> [1] Hu, X., Cheng, L. Y., Chiu, M. H., & Paller, K. A. (2020). Promoting memory consolidation during sleep: A meta-analysis of targeted memory reactivation. Psychological bulletin, 146(3), 218.
>
> [2] Liu, Y., Nour, M. M., Schuck, N. W., Behrens, T. E., & Dolan, R. J. (2022). Decoding cognition from spontaneous neural activity. Nature Reviews Neuroscience, 23(4), 204-214.
>
> [3] Lawhern V J, Solon A J, Waytowich N R, et al. EEGNet: a compact convolutional neural network for EEG-based brain–computer interfaces[J]. Journal of neural engineering, 2018, 15(5): 056013.
>
> ## Weakness 4:
> What are the asterisks in Figure 3?
> ## Response:
> Sorry, we apologize for not specifying the meaning of the asterisks in the figure caption. The asterisks indicate statistical significance, as we conducted Paired T-tests for each individual to assess differences between the models. In commonly used statistical conventions, one asterisk corresponds to a significance level (p-value) below 0.05, two asterisks below 0.01, and three asterisks below 0.001. Updated pdf is uploaded.

---

> ### Author Response · Authors · 2023-11-18
> **Response to Reviewer BG5d (2/2)**
>
> ## Weakness 5:
> Even if the data is released, other venues (more focused on health/physiological signals) would be suitable for this paper.
> ## Response:
> We respectfully disagree with this suggestion. Our research, which centers on decoding memory content from brain activity during sleep, has profound implications for the field of neuroscience. Here are the reasons why our study aligns strongly with neuroscience research:
>  - **Memory Consolidation**: A primary focus of our research is on understanding and decoding the neural processes during sleep that contribute to memory consolidation. Sleep is not just a passive state but plays an active role in reorganizing and strengthening memory traces, a process fundamental to learning and memory, which is a core area of interest in neuroscience. The USD provides a groundbreaking approach to decode and understand the neural underpinnings of this process. By aligning neural representations from awake and sleep states, our work provides a way to study how memories are restructured and reinforced during sleep, a subject of great interest in cognitive neuroscience. This aligns with the broader quest in neuroscience to unravel the mechanisms of learning and memory.
>  - **Cognitive Function and Mental Health**: Our findings have significant implications for the study of how sleep influences cognitive functions. The ability of the USD to interpret neural activity during sleep opens avenues for researching the impact of sleep on complex cognitive abilities such as problem-solving, and decision-making. Understanding these relationships is vital for addressing cognitive impairments and enhancing mental health, which are key objectives in neuroscience research.
>  - **Methodological Innovation and Broader Implications**: The development of the USD is a significant methodological breakthrough in neuroscience. Its unique capability to perform cross-subject decoding of neural activity during sleep distinguishes it as a transformative tool, poised for widespread adoption in research related to sleep and memory. Before the introduction of the USD, the field faced a notable limitation: the lack of ability to capture spontaneous memory reactivation during sleep, especially deep sleep, when memory consolidation happens. The implications of this technology extend far beyond its immediate application. The USD is anticipated to significantly advance our understanding of neurological and psychiatric disorders, offering new insights into the cognitive mechanisms of sleep. Additionally, it holds the potential to enhance cognitive functions through a deeper understanding of the neural processes involved in sleep. Perhaps most importantly, it lays the foundation for developing innovative interventions for sleep-related issues, a domain of growing importance in neuroscience.
>
> In conclusion, our study offers substantial contributions to neuroscience, particularly in the realms of memory research, cognitive neuroscience, and research methodology. The innovative nature of our approach and the potential applications of our findings underscore the relevance and importance of this work to the neuroscience community.
>
> ## Details Of Ethics Concerns:
>  - Human subjects research: Data collection process does not have any real details. IRB information is not provided.
>  - Annotation process is not provided. If renumeration or anything else was provided as part of participants/annotators.
> ## Response:
> Sorry for the lack of information. IRB information is provided in **Ethics statement**. Data collection process is provided in Appendix A. Data annotation is provided in Appendix A,B and Fig.5. In this experiment, we don't manually annotate the data; instead, the labels corresponding to the data are automatically aligned by the recording program.

---

### Meta-Review · Area_Chair_ZNzq · 2023-12-14

**Metareview:**

This paper presents a new dataset and first results on semantic sleep segmentation. The dataset is unique, collected for this study, and contains annotated EEG across sleep and awake subjects. The authors demonstrate the ability of their universal sleep decoder to align across subjects and improve sleep accuracy. The reviewers noted these strengths, and in the review process a number of factors were discussed, including lack of baselines, incomplete data availability, mismatch of conclusions with the study capabilities, and general clarity issues related to the details, e.g., of model training and annotation. These points were addressed to varying degrees by the reviewers, however there remains some clarity issues, as well as the incomplete data availability and ambiguity on overall conclusions. Therefore, while the study holds promise, it is likely that a future submission that better addresses these points will fare better.

**Justification For Why Not Higher Score:**

Primary reasons are the the incomplete data availability and ambiguity on overall conclusions. From the discussion with reviewers it seems that the exact phenomenon being studied is not clear, which makes the full assessment of the paper difficult (i.e., how hard is this problem and is the reviewer concerns on low accuracy justified?).

**Justification For Why Not Lower Score:**

N/A

---

### Decision · Program_Chairs · 2024-01-16

Reject